# Human seizures couple across spatial scales through travelling wave dynamics

L.-E. Martinet[1], G. Fiddyment[2], J.R. Madsen[3], E.N. Eskandar[4], W. Truccolo[5,6], U.T. Eden[7], S.S. Cash[1] & M.A. Kramer[7]

Epilepsy—the propensity toward recurrent, unprovoked seizures—is a devastating disease affecting 65 million people worldwide. Understanding and treating this disease remains a challenge, as seizures manifest through mechanisms and features that span spatial and temporal scales. Here we address this challenge through the analysis and modelling of human brain voltage activity recorded simultaneously across microscopic and macroscopic spatial scales. We show that during seizure large-scale neural populations spanning centimetres of cortex coordinate with small neural groups spanning cortical columns, and provide evidence that rapidly propagating waves of activity underlie this increased inter-scale coupling. We develop a corresponding computational model to propose specific mechanisms—namely, the effects of an increased extracellular potassium concentration diffusing in space—that support the observed spatiotemporal dynamics. Understanding the multi-scale, spatiotemporal dynamics of human seizures—and connecting these dynamics to specific biological mechanisms—promises new insights to treat this devastating disease.

[1] Department of Neurology, Massachusetts General Hospital, Boston, Massachusetts 02114, USA. [2] Graduate Program in Neuroscience, Boston University, Boston, Massachusetts 02215, USA. [3] Department of Neurosurgery, Boston Children's Hospital, Harvard Medical School, Boston, Massachusetts 02115, USA. [4] Department of Neurosurgery and Nayef Al-Rodhan Laboratories for Cellular Neurosurgery and Neurosurgical Technology, Massachusetts General Hospital, Boston, Massachusetts 02114, USA. [5] Department of Neuroscience & Institute for Brain Science, Brown University, Providence, Rhode Island 02912, USA. [6] U.S. Department of Veterans Affairs, Center for Neurorestoration and Neurotechnology, Providence, Rhode Island 02912, USA. [7] Department of Mathematics and Statistics, Boston University, Boston, Massachusetts 02215, USA. Correspondence and requests for materials should be addressed to S.S.C. (email: scash@mgh.harvard.edu) or to M.A.K. (email: mak@bu.edu).

Brain dynamics span orders of magnitude in space and time, making population neural activity both rich and difficult to understand. Observing brain activity at each spatial and temporal scale presents unique experimental, logistical and analytical challenges[1]. Moreover, how to optimally assemble and understand these diverse spatiotemporal datasets across spatial scales—especially in behaving humans—remains unknown.

We address this challenge in the specific context of understanding human epileptic seizure, itself a multi-scale phenomenon, spanning microscopic channelopathies to macroscopic clinical manifestations[2,3]. Noninvasive and invasive recording modalities, which are commonly employed clinically[4], reveal the macroscopic features of human brain activity during seizures, including the characteristic rhythms of seizure[5–7] and the coordination between large-scale neocortical networks[8–12]. However, the mechanisms that support this macroscopic activity remain largely unknown. Research into the microscopic dynamics of human seizure provides complementary insights[13,14]. Both *in vivo* microelectrode recordings[15–17] and *in vitro* recordings from resected tissue[18,19] provide detailed dynamic and mechanistic insight into the behaviour of single neurons and small neural populations during human seizure. However, how these phenomena relate to the activity of large scale cortical networks recruited during seizure remains unknown.

One of the most controversial topics in epilepsy is the role of synchronization[20]. Historically, seizures have been considered to reflect a hypersynchronous state[21,22]. However, recent observations at both the macroscopic and microscopic spatial scales suggest that seizures exhibit intervals of both synchronization and desynchronization[23,24]. At the macroscopic spatial scale, voltage activity recorded from distributed brain regions desynchronizes during seizure, and then synchronizes approaching seizure termination[8,9,25]. At the microscopic spatial scale, neurons exhibit heterogeneous firing behaviours at seizure initiation[23]—ahead of the ictal wavefront[26]—and then more coordinated firing later in seizure driven by transient increases in neuronal network spiking rate[27], particularly when spike-and-wave discharges in the local field potential emerge[23,26,28]. A complete, quantitative understanding of the local and large-scale neural network dynamics is essential to characterize the multi-scale phenomena of human seizures, with the potential to improve treatment of epilepsy. Such improvements are essential as little substantial progress in seizure control for pharmacoresistant patients has been made over the past 40–50 years[29,30].

In this manuscript, we investigate how low-frequency (<25 Hz) interactions evolve during human seizures across two spatial recordings scales. To do so, we analyse simultaneous observations of the brain's microscopic dynamics within a spatially restricted area (spanning 4 millimetres or less), combined with broader clinical macroscopic observations (spanning >10 cm). We show that during seizure coupling between these spatial scales increases, and that larger increases in coupling occur at shorter distances. We provide evidence that rapidly propagating waves of activity underlie this increased inter-scale coupling, and that these waves appear consistently for each patient's seizures. We conclude with a computational model that captures the observed seizure dynamics and leads to suggestions of specific mechanisms—namely, the effects of an increased extracellular potassium concentration diffusing in space—that support the observed inter-scale spatiotemporal dynamics of human seizure. Understanding the multi-scale, spatiotemporal dynamics of human seizure—and connecting these dynamics to specific biological mechanisms—promises new insights to treat this devastating disease.

## Results

**Simultaneous observations of seizure across spatial scales.** The multi-scale data consist of cortical voltage observations from two spatial scales. Macroscopic data were recorded from a standard clinical electrocorticogram, configured in an 8-by-8 grid or in series of electrode strips, with an electrode spacing of 1 cm (black circles in the example of Fig. 1a). Microscopic data were recorded from a 10-by-10 microelectrode array (MEA, red in Fig. 1a) with electrode spacing of 0.4 mm. In all cases, the MEA were implanted in the presumptive resective target, outside of the putative seizure onset zone, such that these areas were recruited to seizure. We therefore examine interactions between spatial scales from cortical regions recruited into seizure, and not interactions between the seizure focus and rest of cortex[23,28,31].

We begin in this section with an illustration of the spatiotemporal voltage dynamics at the microscopic and macroscopic scales, and their associations. Example voltage traces from both spatial scales reveal the characteristic temporal features of seizure (Fig. 1b). Following seizure onset, the voltage activity at both spatial scales transitions from low amplitude, fast oscillations to large amplitude spike-and-wave complexes, and seizure termination occurs abruptly[31]. We note that, in this example, seizure onset occurs ~60 s before a visually apparent change in the voltage activity. In this case, seizure onset occurs in another brain region and subsequently recruits the cortical regions observed[32]. Inspection of the multi-scale data during a single spike-and-wave event suggests an organization of activity between the microelectrodes, and between the spatial scales (Fig. 1c); for example, examination of the voltage traces reveals that the spike-and-wave events are temporally delayed between electrodes[25]. To better visualize the spatial organization during the spike-and-wave event, we plot examples of the voltage activity on the MEA and clinical recording array (Fig. 1d). At both spatial scales, spatiotemporal organization appears. At the microscopic scale, a voltage wave sweeps across the entire MEA[25,28,33]. At the macroscopic scale, spatial organization appears present, although less obvious and localized to a subset of electrodes. Careful inspection of Fig. 1d suggests a spatially distributed voltage decrease in the macroscopic data that evolves from the inferior posterior to the superior anterior parts of the temporal lobe (see the grey ellipse on the figure), near the location of the MEA (red circle in Fig. 1a). This example illustrates the phenomenon that motivates this work: to understand the spatiotemporal evolution of brain voltage activity simultaneously occurring across spatial scales during seizure.

To assess the coupling between spatial scales, we use the coherence; a frequency domain measure of linear association, commonly applied to human seizure data[11,12] and neuronal data in general[34,35]. We show an example of the coherence computed between a microscopic and macroscopic electrode pair in Fig. 1e. In this example, times before seizure onset (at time 0 s) and early in the seizure rarely exhibit significant coherence (see Methods). Approximately midway through the seizure (near time 60 s), intervals of significant coherence appear (warm colours in Fig. 1e). These intervals of significant coherence occur at low frequencies (<~12 Hz) and persist until seizure termination. To summarize these results, we compute the average coherence between 1 and 13 Hz. We choose this frequency interval to focus on the low-frequency rhythms that dominate the coherence observed, and avoid non-rhythmic coupling due to slowly changing trends in the data. The example average coherence (Fig. 1f) summarizes the broadband change in coherence between a micro- and macroscopic electrode pair. We note the abrupt increase in the average coherence that persists from mid-seizure (near 60 s) until termination. Repeating this analysis, we compute the average coherence between all microelectrode-to-macroelectrode

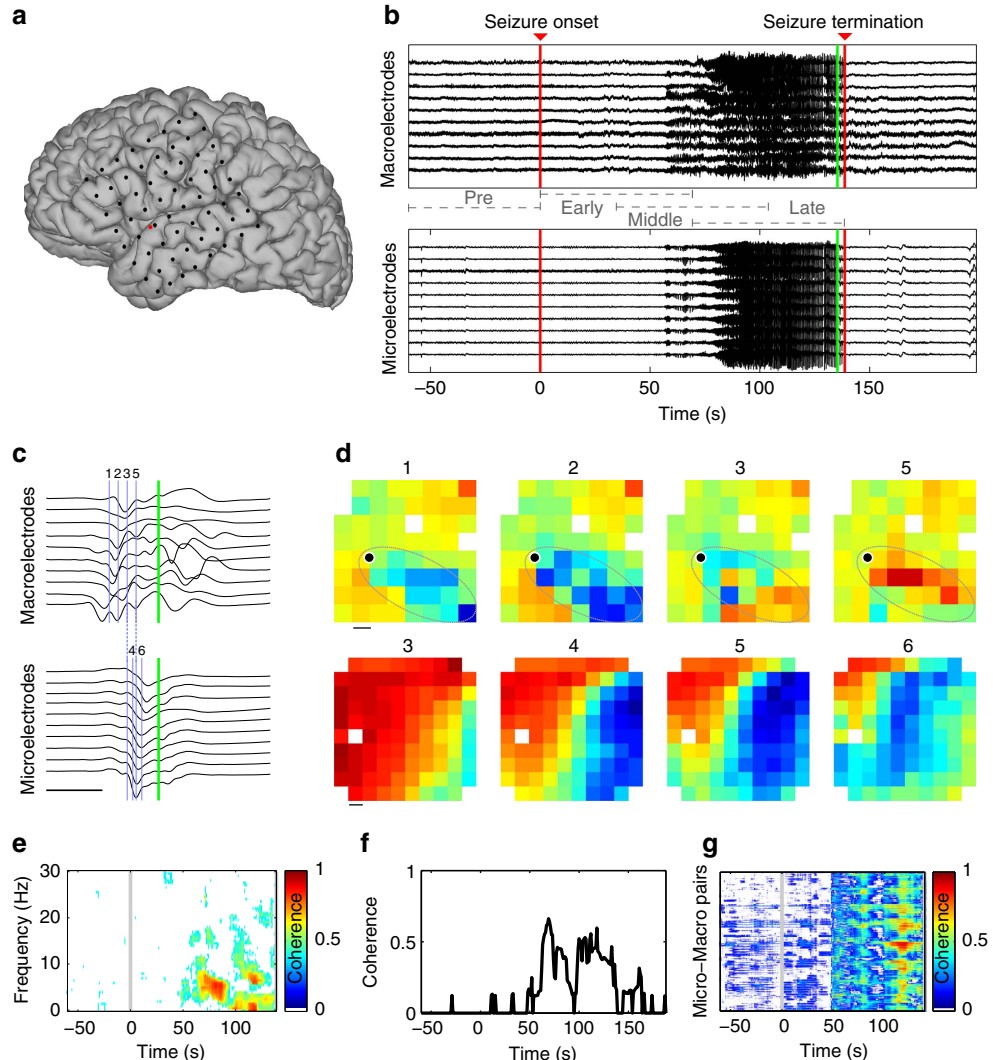

**Figure 1 | Analysis of coherence within and between spatial scales reveals evolution of inter-scale coupling during seizure.** (**a**) Example electrode configuration at the macroscopic and microscopic spatial scales. Each black circle indicates a macroelectrode on the cortical surface. The red circle indicates the location of a 4 mm by 4 mm microelectrode array containing 96 electrodes. (**b,c**) Example voltage traces recorded simultaneously from macroelectrodes (upper) and microelectrodes (lower) during (**b**) seizure and during (**c**) a single spike-and-wave event. The green vertical bars indicate the same time point in both subfigures. The blue vertical bars and labels in **c** correspond to the voltage maps in **d**. Four intervals (pre-seizure, early seizure, middle seizure and late seizure) are indicated in **b**. Scale bar in c indicates 100 ms (**c**). (**d**) Example voltage maps from the macroelectrodes (upper) and microelectrodes (lower) during the spike-and-wave event in (**c**). Warm (cool) colours indicate high (low) voltages. Maps labelled 1, 2, 3 and 5 are spaced by 16 ms, while maps labelled 3, 4, 5 and 6 are spaced by 8 ms (see vertical blue bars in **c**). The approximate area of macroscopic propagation is indicated by a grey ellipse in the top row. Upper scale bar indicates 1 cm, lower scale bar indicates 0.4 mm. (**e**) Example coherogram between a microelectrode and macroelectrode pair. Warm (cool) colours indicate high (low) coherence. Only significant coherence values are shown ($P < 0.005$, not corrected for multiple comparisons, see Methods). (**f,g**) Average coherence from 1 to 13 Hz between (**f**) the microelectrode and macroelectrode pair in (**e**), and (**g**) all microelectrode-to-macroelectrode pairs versus time. In **g**, warm (cool) colours indicate high (low) coherence.

pairs (Fig. 1g). For the example considered here, we find before seizure onset and early in the seizure only weak or insignificant coherence between the spatial scales. Later in the seizure, the inter-scale coherence increases. However, the visualization in Fig. 1g does not indicate how this increase is spatially organized. In what follows, we examine this spatial organization and how it evolves during seizure.

**Coupling increases in seizure and decreases with distance.** In general, observations of brain voltage activity reveal that coupling decreases with increasing distance[36,37]. We examine this observation here in the context of seizure and for coupling across spatial scales. We show an example of the inter-scale coherence

for a single patient and seizure in Fig. 2a. In this figure, each point indicates the average inter-scale coherence between all microelectrodes and each macroelectrode over a window of 10 s, and colour indicates four intervals: pre-seizure (grey), and three seizure intervals (early in pink, middle in red and late in maroon; see legend). Visual inspection of Fig. 2a suggests that inter-scale coherence increases as the seizure progresses, and decreases with increasing distance.

Linear regression of the coherence versus geodesic distance (see Methods) provides three features that summarize these trends in each interval: (1) the left-intercept; (2) the right-intercept, that is, the coherence at maximal distance in the recording; and (3) the slope. We summarize these features for all patients and seizures in Fig. 2b. We find that, consistent with the example in Fig. 2a,

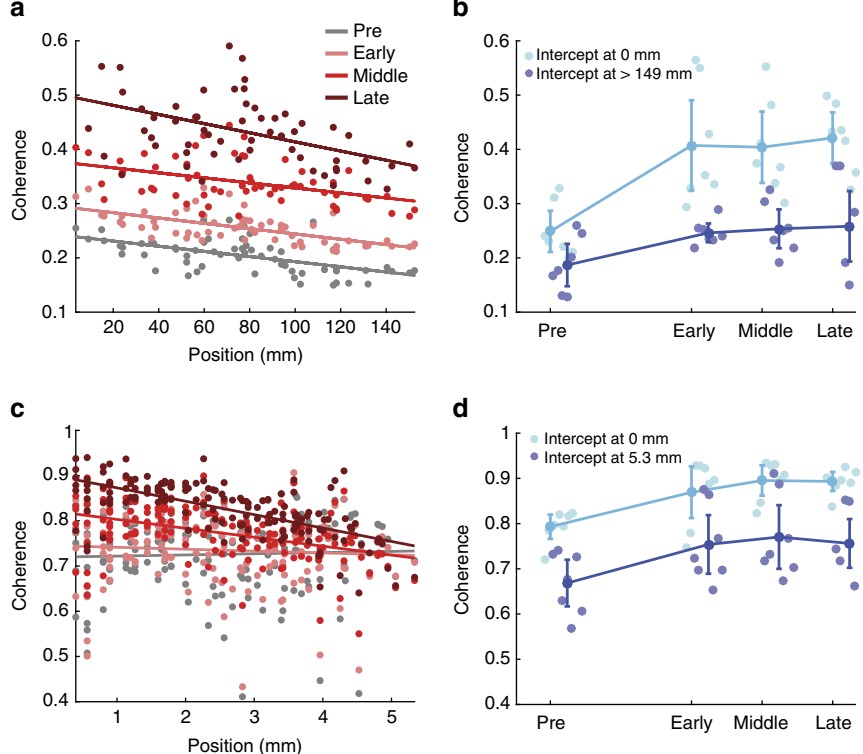

**Figure 2 | Coherence increases during seizure between spatial scales are not spatially uniform.** (**a**) Example average inter-scale coherence (1–13 Hz) between the microelectrodes and macroelectrodes versus distance for a single patient and seizure. Each dot represents the coherence and distance of a macroelectrode during four intervals; grey, pre-seizure; pink, early seizure; red, middle seizure; maroon, late seizure (Fig. 1b). The lines indicate linear regression estimates for each interval. (**b**) Summary of the left-intercepts (light blue) and right-intercepts (dark blue) of the linear regression of inter-scale coherence versus distance for each patient and seizure. Each circle indicates the result for an individual seizure ($n = 7$) in four intervals: pre-seizure, early seizure, middle seizure and late seizure. Circles with vertical lines denote the population mean; error bars indicate two s.e. of the mean. (**c,d**) Same as **a,b** for the coherence between microelectrodes.

the left-intercept and right-intercept increase during seizure compared to pre-seizure ($P = 6e - 5$ and $P = 0.01$ respectively, two-sided $t$-tests, sample size $N_{pre} = 7$ for pre-seizure group and $N_{sz} = 21$ for seizure group), and that the slope decreases ($P = 0.02$, two-sided $t$-test, $N_{pre} = 7$, $N_{sz} = 21$) as seizure progresses. These results indicate that inter-scale coherence increases between the microelectrode and macroelectrode recordings extend over macroscopic distances (up to 15 cm) during the seizure. However, this increase is not spatially uniform. Instead, the inter-scale coherence increase is larger at shorter distances; note that the left-intercept (Fig. 2b, light blue) increases more than the right-intercept (Fig. 2b, dark blue) during seizure and, as expected, the slope is more negative during seizure.

A similar analysis reveals that the coherence increases between the microelectrodes during seizure (Fig. 2c,d). Visual inspection of the coherence versus distance for an example patient and seizure (Fig. 2c) reveals that the coherence is higher between the microelectrodes compared to the inter-scale coherence, and that the coherence tends to increase during seizure. For the population of patients and seizures, we find that the left- and right-intercepts increase significantly during seizure (Fig. 2d, $P = 2e - 4$ and $P = 0.012$ respectively, two-sided $t$-tests, $N_{pre} = 7$, $N_{sz} = 21$), consistent with an overall increase in coherence between microelectrodes during seizure. However, we do not find a significant change in the slope during seizure ($P = 0.98$, two-sided $t$-test, $N_{pre} = 7$, $N_{sz} = 21$); between the microelectrodes, the coherence decreases with distance (that is, the slope is negative), as expected, but this relationship to distance does not change significantly during seizure at this spatial scale.

**Propagating waves organize activity across spatial scales.** We have shown that, during seizures, the coherence increases and that this increase is distance dependent (Fig. 2). To further characterize the spatial organization of this coupling, we use the coherence results between 1 and 13 Hz to estimate the delay between each microelectrode pair, and each micro- and macroelectrode pair (see Methods). We show an example of the estimated delays for all inter-scale electrode pairs in Fig. 3a. In this example, at times preceding seizure onset and early in seizure, we find few significant delays; we note that this lack of significance may result from a lack of significant coherence between electrodes, or lack of a reliable estimate of the delay. For the second half of the seizure we find that the delays span a broad range (in this example, from $-50$ ms to 50 ms) that varies in space and time.

To model the spatial organization of these delays, we perform multiple linear regression of the delay over the two-dimensional (2D) cortical surface. We illustrate this procedure in Fig. 3b,c. Before performing the regression, we first compute the average delay between all microelectrodes and each macroelectrode (circles in Fig. 3b,c). By doing so, each delay estimate reflects the relative delay between a macroelectrode and the MEA. In this way, delays at the macroscopic electrodes are aligned to the microscopic activity; we note that each point in Fig. 3b,c correspond to a single macroelectrode. We then perform multiple linear regression, which corresponds to fitting a 2D plane to these delays. When the estimated delays do not vary linearly in space (example in Fig. 3b), the linear fit is poor and not significantly better than a plane passing through the estimated mean delay

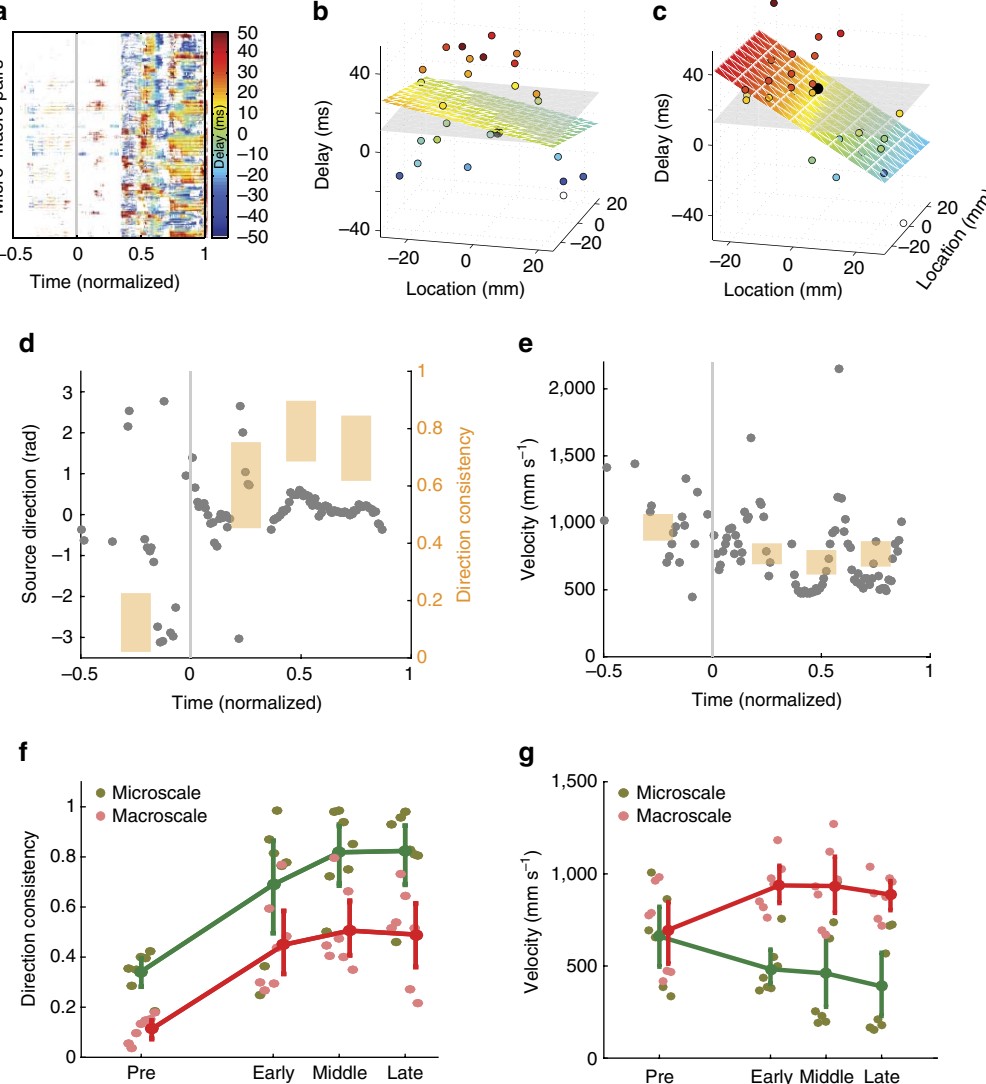

**Figure 3 | Travelling waves of activity propagate within each spatial scale during seizure.** (**a**) Example of the delays between all pairs of microelectrodes and macroelectrodes versus normalized time. Positive (negative) delays are indicated with warm (cool) colours. Intervals that lack significant coherence or fit are white (see Methods). Seizure onset begins at time 0 and ends at time 1. (**b,c**) Examples of robust multiple linear regression of the delay values (circles, vertical axis) versus position on the macroelectrode array. The fit plane is indicated in colour, and the null model in grey. Example of a poorly fit (**b**) and well fit (**c**) spatial distribution of delays. (**d,e**) The estimated (**d**) source direction and (**e**) velocity deduced from the multiple linear regression fit versus time for an example seizure from one patient. Each dot indicates the estimate at a moment in time; a time without a dot indicates that a significant value (see Methods) was not found. The four shaded bars are centered at the mean direction consistency (**d**, right vertical axis), or mean velocity (**f**) estimated in four time intervals of equal size: before the seizure (at negative normalized time) and during three intervals of seizure. The height of each bar indicates the 95% confidence interval. (**f,g**) Population results ($n = 7$) for the (**f**) direction consistency and (**g**) velocity. The direction consistency increases significantly during seizure. The velocity increases significantly during seizure at the macroscale; see Methods.

(the null model, grey surface in Fig. 3b). When the delays do vary approximately linearly in space (example in Fig. 3c), the linear fit captures this spatial organization, and significantly improves upon the null model. We repeat this procedure for data recorded at the microscopic scale by estimating the delays between all microelectrodes and the central microelectrode of the MEA, and then performing the multiple linear regression. We find for the population of patients and seizures a significant increase in the number of plane waves detected during seizures at both the macroscopic ($P = 3e - 5$, two-sided $t$-test, $N_{pre} = 7$, $N_{sz} = 21$) and microscopic spatial scales ($P = 3e - 10$, two-sided $t$-test, $N_{pre} = 7$, $N_{sz} = 21$). This result is consistent with visual inspection of the data (Supplementary Movies 1–4), and the notion that spatial organization of brain activity increases during seizure through the emergence of propagating waves. We note that more plane waves

are detected at the microscopic than macroscopic spatial scale during seizure (mean 45, s.d. 16 at the microscopic spatial scale; mean 31, s.d. 15 at the macroscopic spatial scale), consistent with an increase in local wave propagation—or an increased ability to detect wave propagation—between the spatially restricted micro-electrodes compared to the spatially distributed macroelectrodes.

To further characterize these data, we estimate two quantities from the multiple linear regression: the angle of the wave source and the wave speed (see Methods). We show an example evolution of these quantities for a single patient and seizure in Fig. 3d,e. Before seizure onset and early in the seizure, there are many times at which no estimate of source direction or velocity is available. This lack of estimates may result for a variety of reasons (for example, the inter-scale coherence was not significant, the estimated delay was not significant, the 2D linear regression was

not significant; see Methods). In this way, the data analysis approach is conservative and only includes estimates of phase for which significant evidence occurs throughout multiple analysis steps. Visual inspection suggests that the source direction estimates become less variable approaching seizure termination; we note in Fig. 3d that approximately midway through the seizure (normalized time 0.5) the source direction estimates concentrate just above 0 radians. To characterize this variability, we compute the consistency of the source direction for one pre-seizure interval, and three seizure intervals (bars in Fig. 3d, right vertical axis, see Methods). We find that the direction consistency increases throughout the seizure. Estimates of the velocity remain variable throughout the seizure (Fig. 3e). We conclude that, for the example considered here, the plane waves that appear become more consistent in direction and less so in speed approaching seizure termination.

Repeating these analyses for the population of patients and seizures we find similar results. During seizure, the consistency of the source direction increases at the microscopic and macroscopic spatial scales (Fig. 3f, microscale $P = 2e - 5$, macroscale $P = 8e - 6$, two-sided $t$-test, $N_{pre} = 7$, $N_{sz} = 21$). We note that the direction consistency is higher between the microelectrodes; between macroelectrodes, many features may reduce the consistency of wave propagation, for example inhomogeneity in tissue properties and connectivity, and the folded organization of the neocortex which does not allow us to sample space densely and uniformly. At the spatial scale of the microelectrodes, these variations are less marked, and the approximation of linear wave propagation is more accurate. Observations at both spatial scales provide estimates of wave velocity (Fig. 3g). At the microscale, the velocity varies between a mean value of 660 mm s$^{-1}$ pre-seizure to a mean value of 390 mm s$^{-1}$ at seizure termination, although this difference is not highly significant ($P = 0.051$, two-sided $t$-test, $N_{pre} = 7$, $N_{sz} = 21$), while at the macroscale the velocity increases during seizure (mean value of 696 mm s$^{-1}$ at seizure onset and 888 mm s$^{-1}$ at seizure termination, $P = 0.009$, two-sided $t$-test, $N_{pre} = 7$, $N_{sz} = 21$). These results are qualitatively consistent with other reports of wave speed during human seizures, which vary from ~100 mm s$^{-1}$ to 1,000 mm s$^{-1}$ according to the spatial scale observed, the delay estimation procedure, and the type of distance measure (for example, geodesic distance along the brain folds versus Euclidean distance)[25,28,33].

We have shown that travelling waves of activity appear within each spatial scale during seizure, and that, within each scale, the source directions for these waves become more consistent. We now examine whether these source directions align across spatial scales. We begin by plotting, for each patient, the average source direction during the seizure at each spatial scale (Fig. 4a). As expected, the source direction at the macroscale (red arrows) indicates the location of largest delays (warm colour circles in Fig. 4a). We note that the average source directions at the macroscale (red arrows) and microscale (green arrows) tend to lie within the same quadrant, consistent with travelling waves propagating in the same direction across the two spatial scales. Although the source directions vary between patients, the direction remains similar for each seizure of a patient (Fig. 4b). To characterize the differences in source direction between the two spatial scales, we compute their circular direction difference at each moment in time during seizure. The circular direction differences for each patient concentrate near 0 radians (Fig. 4c). We conclude that, for the three patients, waves of activity propagate in a similar direction across spatial scales, and that for each patient these waves propagate in a similar direction for each seizure. These results suggest that waves observed at the microscale reflect the local effects of macroscopic wave propagation over larger cortical areas.

**A model replicates the spatiotemporal dynamics of seizure.** To propose mechanisms that support the multi-scale interactions observed during seizure we implement a computational model. We chose here a mean-field model, consistent with the spatial scale of the field data observed. Many mean-field formulations exist[38], and here we focus on the formulation originally proposed in refs 39,40, and extended in ref. 41 to simulate the effects of anaesthesia. We choose this formulation because it has been successfully extended and interpreted in numerous way, including to study sleep[42–44], cognitive states[45], the effects of anesthesia[46,47] and seizure[31,48,49]. We consider a model consisting of two cortical cell populations (excitatory and inhibitory) interacting reciprocally through synaptic interactions (Fig. 5a). The inhibitory cell populations are also coupled with gap junctions, consistent with experimental observations[41,50,51]. Both the synaptic and gap junction coupling occur only between spatial neighbours, and no longer distance synaptic interactions are included. This model has been shown to support both travelling waves and temporally fixed spatial patterns (that is, Turing patterns)[41,50].

We update this model to simulate the temporal evolution of seizure by including a slowly evolving variable representing the changing concentration of extracellular potassium, which increases dramatically during seizure and other dysfunctional brain states[52–57]. In the model, activity of either cell population increases the local extracellular potassium concentration, which gradually decays (for example, due to uptake by glial cells) and also diffuses in space. We assume that changes in the local extracellular potassium concentration impact the neuronal dynamics in two ways. First, we assume that an increase in the local extracellular potassium concentration increases the excitability of the local neural populations by increasing the reversal potential for potassium. We model this impact by increasing the resting potential of both cell populations with increasing local extracellular potassium concentration. Second, we assume that increases in the local extracellular potassium concentration act to decrease the inhibitory-to-inhibitory gap junction diffusive-coupling strength. This effect is included to mimic the closing of gap junctions caused by the slow acidification of the extracellular environment late in seizure associated with increased extracellular potassium[58,59] and other sources, such as the accumulation of lactic acid and $CO_2$ (ref. 53).

We simulate seizure initiation on a 30 cm by 30 cm flat cortical surface by activating a 'source' of increased excitability at one spatial location (see Methods). This source maintains increased excitability for 100 s, and then the excitability is reduced to simulate termination of the cortical seizure source. We examine the resulting spatiotemporal dynamics at both simulated micro-electrodes and macroelectrodes (see Methods). Examples of the model's temporal dynamics reveal a transition from pre-seizure inactivity to large activity fluctuations characteristic of seizure (Fig. 5b). Examination of the model's spatial dynamics reveals a distinct sequence of patterns. Before activation of the wave source, the model exists in a 'healthy' state in which cortical activity is small. Upon activation of the wave source, excitability spreads from the wave source over the surface and an approximately static spatial pattern emerges, consisting of active and inactive regions (Fig. 5c, label i)[41,50]. During this interval, the initial focus of activity is not clearly distinguished in the 30 cm by 30 cm plane. As time evolves, the extracellular potassium concentration increases in the active regions, thereby increasing the excitability and reducing the gap junction strength in these regions (Supplementary Figure 1). Eventually, these slow changes induce a transition to an interval of local wave propagation, as many brain regions initiate waves that emerge and collide (Fig. 5c, label ii). We note that active regions in the approximately static spatial patterns act as transient, secondary sources for this

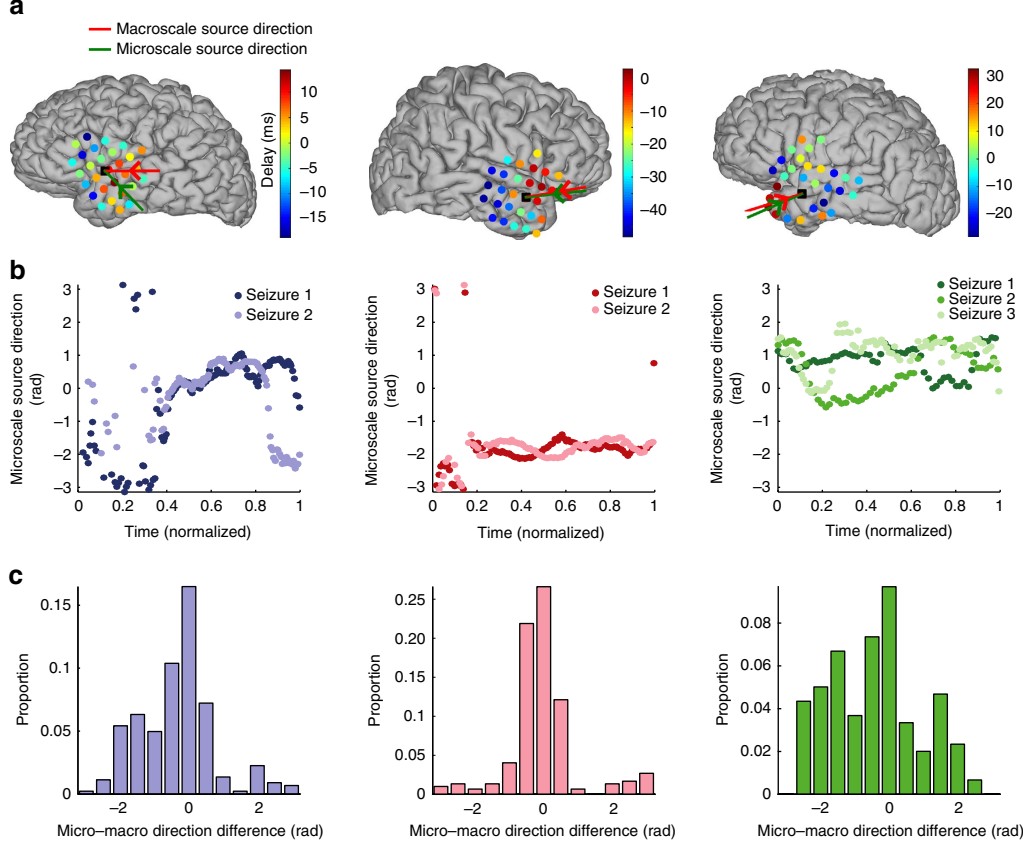

**Figure 4 | Waves propagate in similar directions across a patient's seizures and between spatial scales.** (**a**) Average delay at each macroelectrode (circles) and source direction at the macroscale (red arrow) and microscale (green arrow) for each patient during a single seizure. Delays range from 32 ms to − 48 ms. The location of the MEA is indicated by a black square. (**b**) The source directions at the microscale versus (normalized) time are similar for each seizure (colour) of the three patients, but not between patients. (**c**) The distribution of circular direction differences between the source directions at the microscale and macroscale during seizure concentrate near 0.

local wave propagation. As the extracellular potassium continues to spread, these waves become more spatially organized until the initial source becomes the clear origination point of all waves, and all transient secondary sources of local wave propagation vanish (Fig. 5c, label iii). In this model, the slow evolution of the extracellular potassium concentration navigates the dynamics between different spatiotemporal stages.

Applying the analyses used to study the *in vivo* activity to the simulated data, we find qualitatively consistent results. The coherence increases during seizure between microelectrodes ($P < 5e − 5$ for both the left- and right-intercepts, two-sided $t$-tests, $N_{pre} = 10$, $N_{sz} = 30$) and between spatial scales ($P < 1e − 4$ for both the left and right-intercepts, two-sided $t$-tests, $N_{pre} = 10$, $N_{sz} = 30$, Fig. 5d). The direction consistency increases during seizure at both spatial scales ($P < 5e − 3$, two-sided $t$-test, $N_{pre} = 10$, $N_{sz} = 30$, Fig. 5e), while the velocities simulated are consistent with the *in vivo* values; both range between 50–400 mm s$^{−1}$ (Fig. 5f). We note that, if instead the source of increased excitability appears at random spatial locations over time, then the direction consistency during seizure is significantly smaller ($P = 0.003$, two-sided $t$-test, $N = 10$ for the model, $N = 7$ for the *in vivo* data) in this model (mean 0.47, s.d. 0.22, during the late seizure interval) compared to the *in vivo* data (mean 0.81, s.d. 0.16 during the late seizure interval; Supplementary Figure 2; Fig. 6b). Finally, the source directions align during seizure (Fig. 5g), consistent with travelling waves that propagate in the same direction across the micro- and macroelectrodes. These

results support the conclusion that an established mean-field model, updated to mimic changes in extracellular potassium dynamics, simulates important features consistent with the human seizure data.

Related scenarios of seizure evolution have been proposed[25,26]. The *in vivo* activity and simulated data described above are consistent with a small territory of increased activity—an ictal core—that produces widely and rapidly distributed low-frequency (2–50 Hz) fields extending well beyond the ictal core, over broad, multilobar regions[26]. These low-frequency fields travel as waves, which consist of fast-moving synaptic potentials, and produce the large amplitude electroencephalogram (EEG) signature of seizures over broad cortical areas. In another scenario, the seizing territory expands as a slowly advancing, sharply demarcated, narrow (< 2 mm) band of multiunit firing, termed the ictal wavefront. Travelling waves arise behind the ictal wavefront as it slowly and radially expands across the cortex. This scenario benefits from both clinical[25] and experimental[60] observations.

Our model provides a framework to simulate and compare an expanding ictal wavefront scenario[25] with the scenario of a spatially restricted, cortical source. To do so, we consider the simplest formulation of an expanding ictal wavefront: a 2D boundary of increased excitation that spreads outward at an approximate speed of 1 mm s$^{−1}$ (ref. 26). This spatial spread includes a random component, so that the ictal wavefront appears as a distorted circle in the 2D plane (Supplementary Methods). This distortion reflects the fact that the ictal wavefront is unlikely

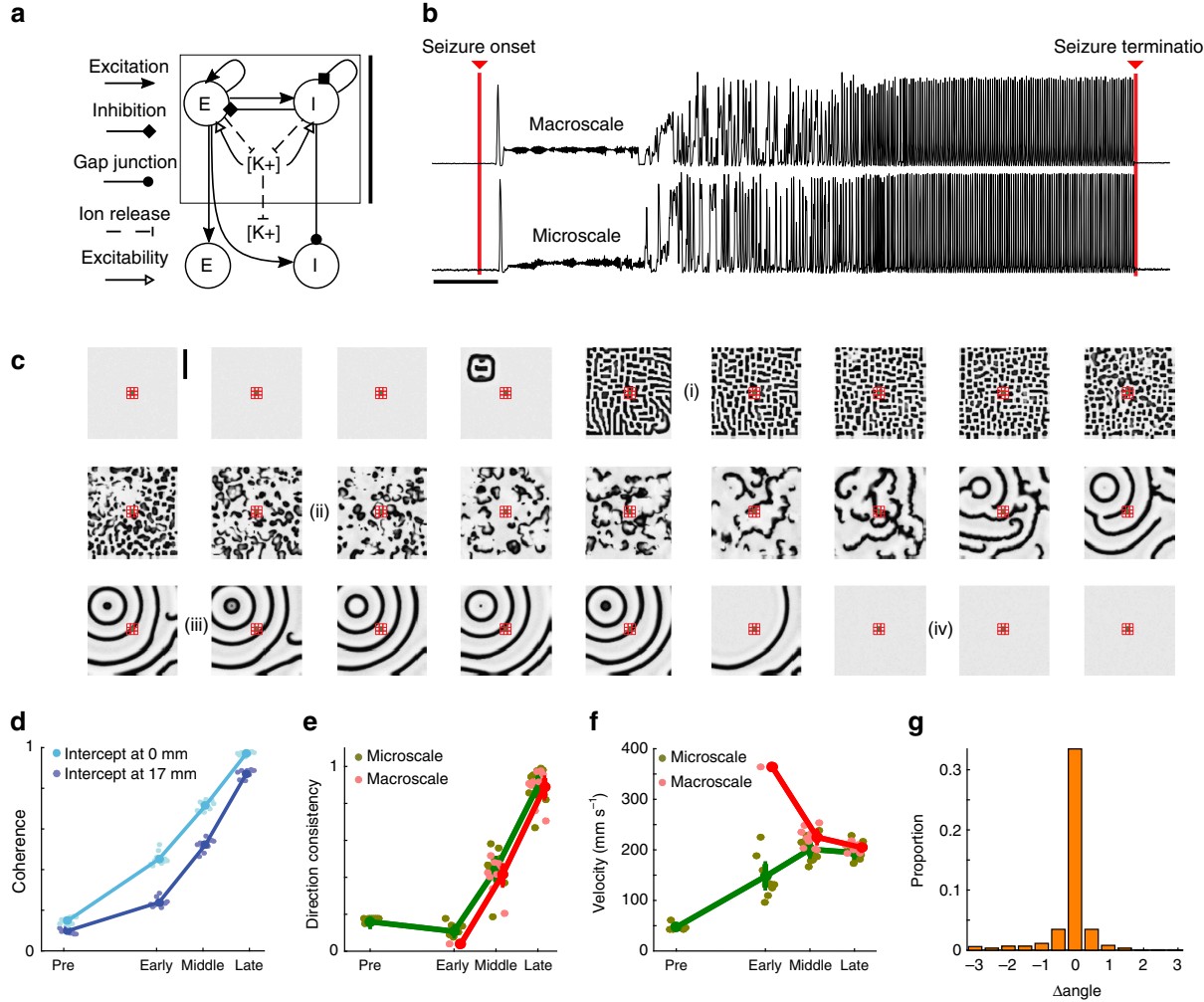

**Figure 5 | A mean-field model of cortical activity and diffusion of extracellular potassium reproduces the spatial-temporal dynamics of human seizure.**
(**a**) Cartoon illustration of the model. Excitatory (*E*) and inhibitory (*I*) neural populations interact through synaptic interactions and gap junctions between spatial neighbours. Activity of either cell population increases the local extracellular potassium concentration, which diffuses in space. Vertical scale bar on the right of the panel indicates 3 mm. (**b**) Example traces of simulated activity at a macroelectrode (upper row) and microelectrode (lower row). The red bars labelled 'Seizure onset' and 'Seizure termination' indicate the time when the excitability of the cortical source was increased and decreased, respectively (see Methods). Scale bar indicates 10 s. (**c**) Example spatial maps of simulated activity. Each subfigure shows a snapshot of the excitatory population activity (white 0 Hz, black 25 Hz) on the 30 cm by 30 cm surface; the time between subfigures is 5 s and time progresses from left to right, top to bottom. The cortical source (visible near the upper left corner in the fourth subfigure) ignites the activity. A static mosaic pattern (i) then appears, followed by spatially local propagation (ii) and concluding in travelling waves driven by the seizure source (iii). When the source is inactivated (iv), propagation ceases. The simulated microelectrodes (green) and macroelectrodes (red) are indicated near the center of each map. Vertical scale bar in top left panel indicates 10 cm. (**d**–**g**) Simulated dynamics at each scale produce results consistent with the *in vivo* data. (**d**) A linear fit of the coherence versus distance reveals an increase in the left and right intercepts during seizure. (**e**) The direction consistency increases during seizure and (**f**) the velocity approaches values between 100 and 300 mm s$^{-1}$ during seizure. (**g**) The difference in source direction between spatial scales concentrates at 0 radians. In all figures, mean and s.e. of the mean computed as in Figs 2 and 3, with $n = 10$.

to spread as a perfect radial wave across different cortical areas, gyri and sulci. Behind this expanding wavefront, in the seizing territory, travelling waves emerge consistent with the classic EEG signature of seizures (Fig. 6a). Computing the direction consistency approaching seizure termination for the micro-electrode data simulated in this model, we find significantly smaller values (mean 0.40, s.d. 0.11) compared to the *in vivo* data (mean 0.81, s.d. 0.16; $P = 2e - 5$, two-sided *t*-test $N = 10$ for the model, $N = 7$ for the *in vivo* data) and compared to the model with a fixed cortical source (mean 0.89, s.d. 0.1; $P = 5e - 9$, two-sided *t*-test, $N = 10$ in both groups; Fig. 6b). In this simulation, in which the entire ictal wavefront remains active, different locations on the expanding ictal wavefront emit travelling waves, which then propagate differently across the microelectrode array over

time, thus reducing the direction consistency (Fig. 6c). We do not find a significant difference ($P = 0.32$, two-sided *t*-test, $N = 10$ for the model, $N = 7$ for the *in vivo* data) between the direction consistency computed for the *in vivo* data and the model with a fixed cortical source. In this scenario, the fixed spatial location of the cortical source results in waves that travel consistently across the microelectrode array (Fig. 6d). However, we note that, if only a small region of the ictal expanding wavefront remained active before seizure termination, then this region could also act as a slowly drifting source of cortical waves.

## Discussion
In this manuscript, we showed that coherence increases within and between spatial scales during seizure. We proposed a

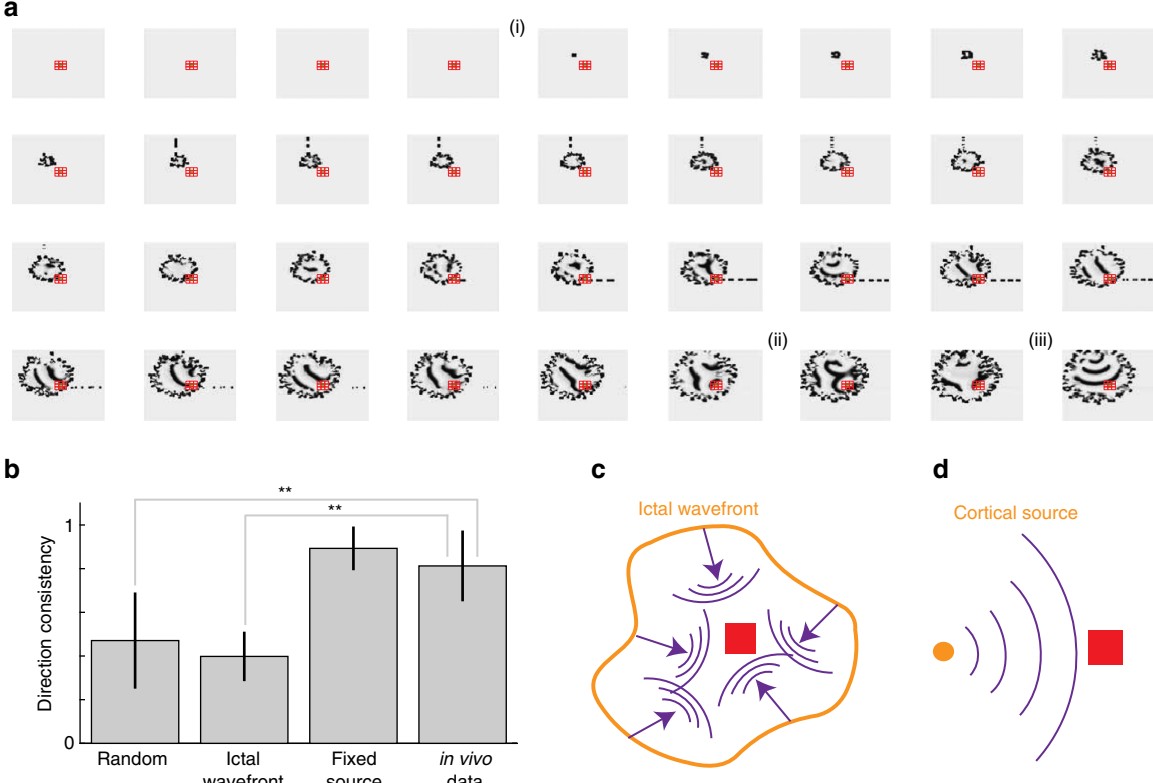

**Figure 6 | Simulations of an expanding ictal wavefront produce direction consistency measures inconsistent with the *in vivo* data.** (**a**) Example spatial maps of simulated activity for a simple expanding ictal wavefront scenario. The arrangement and colour scale are the same as in Fig. 5a. An ictal wavefront emerges (i) and slowly recruits cortical territory. As the ictal wavefront expands, travelling waves propagate into the recruited territory from different directions; compare (ii) and (iii). (**b**) The direction consistency during the last half of seizure in three simulation scenarios and for the *in vivo* data. Compared to the *in vivo* data, the direction consistency is significantly lower during the second half of seizure for the random source locations and expanding ictal wavefront simulations; **$P < 0.005$, two-sided *t*-test. (**c,d**) Schematic representations for two related scenarios of cortical wave activity during seizure. In **c**, the expanding ictal wavefront (orange) evolves in space to produce travelling waves (purple) that propagate to the microelectrode array (red) from different directions. In **d**, a cortical source (orange) produces waves (purple) that impact the microelectrode array (red) from the same direction.

dynamical reason for this increase: the emergence of propagating waves of activity that sweep across spatial scales. We showed that these waves become more consistent over the course of seizure, and propagate in the same direction across spatial scales for a patient's seizures. We implemented a corresponding computational model of a 2D cortical field to propose that the slow aggregation and diffusion of extracellular potassium supports the observed spatiotemporal dynamics.

The seizing human cortex provides an ideal system in which to study the spatiotemporal dynamics of multi-scale brain activity. A seizure is a stereotyped multi-scale dysfunction proposed to initiate at a (microscopic) source and—in these patients— subsequently recruit all or large portions of the entire observable cortex. To understand the multi-scale, spatiotemporal dynamics of seizure requires assessment of the temporal and spatial organization of simultaneous macroscopic and microscopic voltage recordings from human cortex. These invasive observations, which involve implantation of a subdural clinical macroelectrode array and high-density microelectrode array, are only performed in human patients with refractory epilepsy. Other studies have investigated features of macroelectrode and microelectrode data recorded simultaneously[25,31,61–63], but have not directly assessed the coherence between these two spatial scales.

Many studies have shown that coupling (or functional connectivity) within the macroscopic cortical network increases approaching seizure termination[9,20,64,65]. This coupling increase

has been interpreted as reflecting the seizure onset zone gradually recruiting—and becoming more functionally connected with—the rest of the brain. However, the mechanisms of this increased coupling are not known. This work supports a dynamical understanding of increased functional connectivity: the emergence of propagating waves over the cortical surface[25]. These waves may arise, for example, from a cortical source, which may differ from the seizure focus and may be driven by an unobserved subcortical source. From this cortical source, repeated waves of activity emerge that act to couple the voltage activity recorded from cortical macroelectrode pairs. As the seizure progresses, wave emissions from the cortical source become more salient and act to further entrain cortical activity.

This dynamical understanding is consistent with a computational model in which travelling waves emerge from a cortical source. To simulate a seizure in the model, we increase the activity of the cortical source. Initially, this source is difficult to identify, as a complex pattern of activation and inactivation appears over the simulated cortical surface. For this cortical source to emerge spontaneously requires the slow accumulation and diffusion of extracellular potassium in the model. Dramatic changes in many extracellular ions occur during seizure[53], including increases in extracellular potassium concentration ($[K^+]_o$), which impact neural dynamic and have been proposed as important to seizure activity[52]. The field model proposed here implements these existing concepts, as well as the

observation that increases in $[K^+]_o$ act to close gap junctions, through an acidification of the extracellular environment. As the simulated seizure progresses, the increased excitability of the neural populations and the reduced coordination of inhibitory cells through a loss of gap junctions, supports the emergence of travelling waves. Here the hypothesized role of $[K^+]_o$ could not be tested directly in the human patients; future research that incorporates clinically approved methods to detect extracellular ion concentrations would facilitate such a test. We note that gap junctions between excitatory cells have been proposed as essential to support the fast rhythms associated with seizure (that is, high frequency oscillations or HFO at seizure onset[7]). We did not model these fast rhythms here. Instead, we examined slower seizure rhythms ($<25$ Hz) using a model with gap junction coupling between inhibitory neurons[41,51]. Extending this field model to include faster rhythms would require the inclusion of additional mechanisms.

Although the analysis and modelling provide insight into the multi-scale dynamics of seizure, three important limitations remain. First, by choosing to estimate the spatial organization of the delays using multiple linear regression, we focused on delays linearly organized in space. In doing so, we only captured plane wave activity. The importance of these waves are motivated by visual inspection of the data (for example, Fig. 1), and previous analysis of plane wave propagation[25,28,33]. Understanding the more complex spatiotemporal patterns (for example, spirals[66]) that appear during seizure requires the development and application of additional approaches[67]. Second, the computational model implements numerous simplifications. The cortex is not a 2D sheet with uniform connectivity, homogenous parameters and only two cell populations. We induce a sequence of spatiotemporal patterns in the model through a slow change in a variable representative of the concentration of extracellular potassium; slow changes in other model variables may produce similar sequences. We choose to focus on the concentration of extracellular potassium because changes in this ion concentration have been proposed as an important component of seizure[52–57]. Updating the model to address these limitations and incorporate single neuron activity with changes in extracellular ion dynamics[56,57] would provide additional insight. Third, we analysed here two scales of field activity. Incorporating single unit activity would provide an additional spatial scale, although doing so remains controversial[23,68].

Numerous similarities exist between the results presented in refs 25,26 and those presented here. In both cases, travelling waves are observed with similar speeds that propagate in preferred directions consistently across the macroelectrode and microelectrode domains. These similarities occur despite the small number of subjects analysed (three in ref. 25 and three here, which limits general conclusions) and the different data analysis approaches employed (for example, characterization of ictal discharges in ref. 25 and the entire field time series here). In addition, conceptual similarities link the scenario proposed in refs 25,26 and the one proposed here. Both scenarios suggest that a small cortical source projects travelling waves over a broad cortical area, that these travelling waves induce synchronization in the low-frequency field activity, and that a sufficient dissipation of the cortical source causes seizure termination. However, the two scenarios suggest a different source of ictal activity: in ref. 25 a slowly migrating ictal wavefront is proposed as the source, while here we hypothesize that a fixed cortical location is the source. Using simulations that capture the basic features of these two scenarios (Figs 5 and 6), we find that the fixed cortical source produces propagating waves with a direction consistency similar to the human data analysed in this study, while a uniformly active, expanding ictal wavefront produces a significantly smaller direction consistency.

In the future, two procedures may help further distinguish these two proposed scenarios. First, in the fixed cortical source model, we hypothesize that stimulation delivered to a single cortical location—the cortical source or ictal core—late in seizure will disrupt travelling wave propagation. Alternatively, we expect that single-site stimulation would not disrupt travelling waves that propagate from a uniformly active, expanding ictal wavefront; in this scenario, stimulation would disrupt only a part of the ictal wavefront, while the rest of the ictal wavefront would continue to emit travelling waves into the recruited brain region. Second, in the cortical source model, travelling waves propagate outward from the cortical source, and these waves become more salient approaching seizure termination. Therefore, we hypothesize that identifying the cortical source of these travelling waves—even late in seizure—isolates a potential treatment target, either the cortical area itself or the subcortical areas that drive it. In the expanding ictal wavefront model, the travelling waves are less informative for identifying a treatment target; these travelling waves emerge from the ictal wavefront, which by the end of seizure has propagated away from its point of emergence on the cortex. Instead, tracking the slow evolution of the ictal wavefront may identify a candidate focal treatment target. We note that these two procedures, which exceed the immediate scope of the current study, may reveal that both scenarios occur in a heterogeneous patient cohort[69].

We conclude that the two scenarios, both of which are compatible with many aspects of the observed seizure activity, possess particular distinguishing features in terms of their mechanisms, dynamics and response to stimulation[69]. The two scenarios also differ in the mechanisms of seizure termination. To end seizure abruptly across a wide cortical region, some mechanism must weaken simultaneously the entire spatially distributed ictal wavefront. For example, the entire boundary of the expanding ictal wavefront may encroach on a surrounding area with superior inhibitory restraint. Without this simultaneous cessation, some regions of the ictal wavefront would continue to broadcast travelling waves into recruited cortex. We note that a non-uniform collapse of an expanding ictal wavefront could increase the consistency of travelling waves before seizure termination; for example, if only a small region of the ictal wavefront remained active, then only this region would broadcast cortical waves, which would propagate from a single direction over the brain. Alternatively, in the fixed cortical source model, mechanisms that operate over a limited cortical region could inactivate the cortical source and terminate seizure.

We may interpret these results presented here to suggest two categories of seizure therapy. First, we propose targeting the mechanisms that support seizure. Motivated by the computational model, potential therapies could target the accumulation of extracellular potassium, for example by increasing glial uptake or developing a physical collection mechanism, or act to preserve inhibitory gap junctions. Both modifications prevent the emergence of travelling waves in the model (Supplementary Figure 3). Second, as described above, we propose targeting the source of travelling waves that emerge during seizure. Motivated by the in vivo data and computational model, potential therapies could target the cortical source—rather than the seizure source, which may be inaccessible; firewall the cortical source[70]; or disrupt wave propagation in some way to prevent waves from reaching eloquent cortex.

A unified view of brain activity spans spatial and temporal scales, recognizing that the nervous system consists of interacting molecules, cells and circuits. To achieve this vision requires the collection and analysis of multi-scale brain data, which presents

numerous practical and theoretical challenges. An even greater challenge is to link these multi-scale observations to human function and dysfunction. In this study, we examined multi-scale voltage data collected from human patients during seizure. Analysis of these data identified the dynamics—propagating waves—that link activity across the microscopic and macroscopic spatial scales. A corresponding computational model provided a mechanistic implementation of these human multi-scale voltage dynamics, with testable hypothesis at the microscale (for example, the role of diffusion of extracellular potassium). In this way, the combination of clinical human data with computational models acts to link human behaviour (in this case, seizures) to molecules.

## Methods

**Patients and recordings.** Three patients (males ages 45, 32 and 21 years) with medically intractable focal epilepsy underwent clinically indicated intracranial cortical recordings using grid electrodes for epilepsy monitoring. Clinical electrode implantation, positioning, duration of recordings and medication schedules were based purely on clinical need as judged by an independent team of physicians. Patients were implanted with intracranial subdural grids, strips and/or depth electrodes (Adtech Medical Instrument Corporation) for 5–14 days in a specialized hospital setting and continuous intracranial EEG were performed (500 Hz sampling rate). The reference was a strip of electrodes placed outside the dura and facing the skull at a region remote from the other grid and strip electrodes. One to four electrodes were selected from this reference strip and connected to the reference channel.

These patients were also implanted with an additional $10 \times 10$ (4 mm × 4 mm) NeuroPort microelectrode array (MEA; Blackrock Microsystems, UT) in a neocortical area expected to be resected with high probability, in either the middle or superior temporal gyrus. The MEA consist of 96 recording platinum-tipped silicon probes, with a length of either 1 or 1.5 mm, corresponding to neocortical layer III as confirmed by histology after resection. The reference was an electrode placed either subdurally or epidurally and remote from the recording site[23]. The macroelectrode and MEA data sets were aligned using a pulse-coded signal delivered simultaneously to both recording systems. Signals from the MEA were acquired continuously at 30 kHz per channel and then subsequently down-sampled to 500 Hz by low-pass filtering (zero-phase forward and reverse finite impulse response filter of order 1,000 with cutoff frequency of 250 Hz) and interpolating voltages at the same time-points as the intracranial EEG.

Seizure onset times were determined by an experienced encephalographer (S.S.C.) through inspection of the macroelectrode recordings, referral to the clinical report and clinical manifestations recorded on video. The seizure end time was defined as the latest time at which both the microelectrode and macroelectrode recordings displayed large amplitude ictal activity. The number of seizures varied across the participants. Owing to operational issues, not all of these seizures were recorded or provided data with a high signal-to-noise ratio. We selected 7 seizures among the three participants. Seizure onsets were detected approximately 2–3 cm away from the MEA, based on the clinical macroelectrodes. These recordings were therefore outside the seizure onset zone.

These data have been previously used in other studies[23,27,31,33]. The three patients analysed here correspond to patients P1, P2 and P4 in study[28]. For a detailed clinical summary of each patient, see Patients P1, P2 and P4 of[28].

All patients were enrolled after informed consent was obtained and approval was granted by local Institutional Review Boards at Massachusetts General Hospital and Brigham Women's Hospitals (Partners Human Research Committee), and at Boston University according to National Institutes of Health guidelines.

**Coherence and delay estimation.** We estimated the time-dependent coherence and its phase between pairs of electrodes using the multi-taper method implemented in the Chronux Toolbox for MATLAB[71]. Electrode pairs consisted of either two microelectrodes, or one microelectrode and one macroelectrode. We divided the data into 10 s windows with 9 s overlap, beginning 60 s before seizure onset and ending at seizure termination and computed the coherence within each window using a time-bandwidth product of 20 (bandwidth of 2 Hz) and 39 tapers, which is chosen to be one less than the Shannon number (or one less than twice the time-bandwidth product)[72]. We declared significant the values of the coherence larger than the theoretical confidence level at 99.5%. Repeating the analyses using a smaller window of 2 s and a time-bandwidth product of 4 lead to qualitatively similar results. We note that the interpretation of coherence, or any coupling measure, from referenced or re-referenced data requires care[73]. Here we chose not to re-reference the data. Because we compute the coherence between many electrode pairs across spatial scales with a different physical reference at each spatial scale, and then analyse the spatial organization of the delays inferred from this coherence, we would expect to find no evidence of travelling waves if the coherence results were dominated by spurious effects[74]. However, we do find evidence for travelling waves, which suggests that these results are not dominated

by a reference signal. We also note that the model data, which is reference-free, produces dynamics consistent with the *in vivo* data; this consistency supports a relatively quiet *in vivo* reference.

We estimated the delay between electrode pairs using the phase of the coherence following the approach developed in ref.12. We considered the (1–13 Hz) frequency range, and identified an interval of 3 Hz or larger with consecutive significant coherence. From this interval we computed a linear fit of the phase of the coherence versus frequency. Whenever the interval existed, and the fit was significantly better than a constant term model ($P < 0.05$ for the F-test on the regression model), its slope provided an estimate of the group delay between the electrode pair[12]. Otherwise, the delay between the electrode pair was considered undefined.

We investigated the relationship of the distance between an electrode pair and its coherence using a linear fit. The distance between electrodes within the microelectrode array was computed using a spacing of 0.4 mm between neighbouring electrodes. To compute the distance between a macroelectrode and the microelectrode array, we used the co-registered three-dimensional (3D) positions of the macroelectrode on the brain surface (see ref. 75 and section Anatomical figures) to estimate the geodesic distance. Repeating the analysis of coherence versus distance (Fig. 2) using the macroelectrode grid spacing of 1 cm, we find qualitatively similar results.

To identify waves travelling within the microelectrode array during each window, we performed a linear regression of the delays **D** as a function of the 2D microelectrode positions (**X,Y**) such that $\mathbf{D} = b_0 + b_1\mathbf{X} + b_2\mathbf{Y}$, where $b_0$, $b_1$ and $b_2$ are the parameters to estimate. Here we consider delays at each microelectrode relative to the microelectrode at the center of the MEA. Two criteria were required for the 2D fit to be considered valid: (i) At least 50% of the electrodes for that time window must have a defined delay; (ii) The fit must be significantly better than a constant term model ($P < 0.05$, F-test that the two slope estimates are both 0). When a wave is identified (that is, a valid fit), two statistics were computed from the fit: the wave speed estimated as $\frac{1}{\sqrt{b_1^2 + b_2^2}}$; and the direction of the wave source, estimated as the four-quadrant inverse tangent, with horizontal coordinate $b_1$ and vertical coordinate $b_2$. To quantify the consistency of the wave source directions $\theta_j$ in a given interval containing N direction estimates, we compute the phase locking value $\phi$ defined as $\phi = \frac{1}{N}\left|\sum_{j=1}^{N} e^{-i\theta_j}\right|$; see ref. 76. We refer to the result here as the 'direction consistency' to avoid confusion with the phase estimated from the coherence. We perform a similar procedure for the macroelectrode data. The delay between the MEA and a single macroelectrode was computed as the mean delay between all microelectrodes and the macroelectrode. We considered the 30 macroelectrodes closest to the MEA, and electrode positions on the brain surface were estimated through coregistration (see section Anatomical Figures).

For most measures, we computed the mean and its 95% confidence interval averaged over four windows representative of relevant time intervals during seizure. We define these windows as follows, using normalized seizure duration, so that the seizure begins at time 0 and ends at time 1: the pre-seizure window (label 'Pre' in figures) from $-0.5$ to 0, the early seizure window (label 'Early') from 0 to 0.5, the middle seizure window (label 'Middle') from 0.25 to 0.75, and the late seizure window (label 'Late') from 0.5 to 1.0. Results in the text and in the figures represent the estimated grand mean $\pm$ 95% confidence interval of the described statistics, computed across seizures. The confidence intervals were estimated with a bootstrap resampling procedure using 1,000 samples with replacement. To test for differences between the pre-seizure period and seizure, we used the *t*-test and report the *P* values without correction in the manuscript.

**Anatomical figures.** To create anatomical representations of electrode placement and seizure spread (Figs 1 and 4), we used Freesurfer[77] to reconstruct a 3D model of the cortical surface of each patient using preoperative high-resolution magnetic resonance imaging data. We then co-registered these magnetic resonance imaging data with a postoperative computed tomography scan showing the location of the intracranial electrodes to obtain the coordinates of electrodes in the space of the reconstructed 3D model of the cortex. These procedures are described in detail in ref. 75.

**Computational model.** We implemented an extension of the mean-field model originally proposed in ref. 41. The original implementation of this model, including definitions of all variables and parameters, and MATLAB code can be found in the Supplemental Material of ref. 41. We simulated this model here with the following three modifications: (i) We used no flux boundary conditions. (ii) We included a model of depolarization block, so that the firing rate of each population approaches zero as the voltage exceeds $-20$ mV; that is, we apply a Gaussian activation function, rather than the standard sigmoid function[78]. Some recent observations from human seizing cortex suggest an important role for depolarization block[78,79], while others suggest this role may depend on the type of seizure[23,80]. We do not find a critical role for depolarization block in the large amplitude, low-frequency dynamics simulated here. However, to simulate the low amplitude, high-frequency rhythms commonly observed at seizure onset would require inclusion of additional mechanisms, for example an additional cell population, in which depolarization block may serve an important role. (iii) We simulated extracellular potassium

dynamics at each location that obey the following differential equation:

$$\frac{dK}{dt} = -\delta K + \frac{C_1(Q_e + Q_i)}{1 + e^{-(Q_e + Q_i) - 15}} + C_2 \nabla^2 K \qquad (1)$$

where $K$ represents a (unitless) proportion of extracellular potassium. The first term in (1) represents a decay (rate $\delta = 0.1\,s^{-1}$) of $K$. The second term acts to increase $K$ when either population (with excitatory firing rate $Q_e$ and inhibitory firing rate $Q_i$) at the location is active with scale factor $C_1 = 0.15$. The third term represents diffusion of $K$ (diffusion coefficient $C_2 = 1\,cm^2\,s^{-1}$) between neighbouring spatial locations; here $\nabla^2$ is the 2D Laplacian operator.

The proportion of extracellular potassium $K$ acts to decrease the gap junctions between inhibitory populations ($D_{ii}$), and increase the resting voltages of the excitatory ($\Delta V_e^{rest}$) and inhibitory ($\Delta V_i^{rest}$) populations through the following differential equations:

$$\begin{aligned} \frac{dD_{ii}}{dt} &= -\frac{D_{ii}}{\tau_D} \\ \frac{d\Delta V_b^{rest}}{dt} &= \frac{\Delta V_b^{rest}}{\tau_V} \end{aligned} \qquad (2)$$

where $b = \{e,i\}$, and we set the time constants so that these variables change slowly: $\tau_D = 4\,s$ and $\tau_V = 25\,s$.

All of the model parameters are identical to the original model in Table I of ref. 41 except for: the neuron time constants (here $\tau_{e,i} = 0.02\,s$ versus 0.04 s), the initial offset to the resting potential (here $\Delta V_{e,i}^{rest} = 2.5, 0.1\,mV$ versus 1.5, 0 mV), the tonic excitatory flux entering from subcortex (here $\langle \phi_{eb}^{sc} \rangle = 150\,s^{-1}$ versus $300\,s^{-1}$), the subcortical noise scale factor (here 2 versus 4), and the axonal conduction speed (here $280\,cm\,s^{-1}$ versus $140\,cm\,s^{-1}$).

We simulated the model on a 100-by-100 square grid, corresponding to a 300 mm by 300 mm cortical surface. Each position on the grid represents a coarse grained approximation to cortex over an approximately $9\,mm^2$ area[41]. On this surface, we included a 'cortical source' of increased excitability at position $(x,y) = (75\,mm, 75\,mm)$; we modelled this excitability as a threefold increase in the resting membrane voltage of the excitatory populations at this location. We examined the simulated spatiotemporal dynamics at two spatial scales. We considered a 3-by-3 'microelectrode array' centered in the cortical region (Fig. 5c). Each microelectrode recording corresponded to the excitatory population activity at the selected location (representative of a $9\,mm^2$ area of cortex). Surrounding this microelectrode array, we placed nine 'macroelectrodes'. Each macroelectrode recording corresponded to the average excitatory population activity from a $108\,mm^2$ area of cortex (Fig. 5c). This simple approximation represents the notion that a macroelectrode records the summed activity from multiple cortical columns. At each grid position we simulated the 16 original model variables from ref. 41 and the four additional (slow) variables we introduced in (1) and (2).

We simulated the model dynamics for 180 s, and changed only one parameter during this time: the source activation. For the first 40 s the source was inactive ($\Delta V_e^{rest} = 1\,mV$ at the source), while in the middle 100 s the source was active ($\Delta V_e^{rest} = 3\,mV$ at the source), and in the last 40 s the source was inactive ($\Delta V_e^{rest} = 1.5\,mV$ at the source). The last value of $\Delta V_e^{rest}$ was chosen to be consistent with the increased resting values of the cortical sheet due to the increased concentration of extracellular potassium (variable $K$). The model dynamics evolve with no further intervention; that is, no other fixed parameters are adjusted during the simulation. We repeat the entire simulation 10 times, each with a different noise instantiation, to create the results in Fig. 5.

**Code and data availability.** All analyses and modelling were performed using custom designed algorithms written in MATLAB (MathWorks, Inc). An algorithm to estimate wave properties (direction and speed) from an interval of spatiotemporal data, a 10 s example of microelectrode array data and the model implementation and simulations to reproduce an instance of the results in Fig. 5c are available for re-use or further development at the repository: https://github.com/Mark-Kramer/Seizure-Waves.

**Data availability.** The seizure data that support the findings of this study are available on request from the corresponding author S.S.C. The data are not publicly available due to them containing information that could compromise research participant privacy/consent.

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

## Acknowledgements

M.A.K., L.-E.M., U.T.E. and S.S.C. were supported by the National Institute of Neurological Disorders and Stroke Award R01NS072023. M.A.K. was supported by the National Science Foundation Division of Mathematical Sciences Award Number 1451384. G.F. was supported by a Pre-Doctoral Training Grant from the Epilepsy Foundation Award #330118. W.T. was supported by the National Institute of Neurological Disorders and Stroke Award R01NS079533; the U.S. Department of Veterans Affairs, Merit Review Award I01RX000668; and the Pablo J. Salame '88 Goldman Sachs endowed Assistant Professorship of Computational Neuroscience.

## Author contributions

L.-E.M, W.T, S.S.C. and M.A.K. wrote the manuscript. L.-E.M., U.T.E. and M.A.K. analysed the data. L.-E.M. and M.A.K. did the computational modelling. J.R.M., E.N.E., W.T. and S.S.C. collected the data. L.-E.M., G.F. and M.A.K. aligned the MEA and intracranial EEG data. J.R.M. and E.N.E. implanted the MEAs. All authors contributed in editing the manuscript.

## Additional information

**Competing interests:** The authors declare no competing financial interests.

