## [Peer Review File · Nature Communications]

Reviewers' Comments:

Reviewer #1 (Remarks to the Author)

The manuscript entitled “Human seizures couple across spatial scales through traveling wave dynamics” by Martinet et al., is a well written report on neocortical seizure propagation outside the epileptic focus. The experimental data provide insight into the propagation of seizure activity across two spatial scales in three patients. The presentation of the results is straightforward and easy to interpret, especially since the propagation properties at the micro- and macro-electrode scales are similar. The signal analysis is appropriate and the associated modeling approach is extremely useful and seems to be capable of explaining observations. Overall, I find the data and analyses this paper of high interest to the field, but encourage a revision addressing the points listed below.

1. Line 18: The latest definition of epilepsy (ILAE 2014) does not necessarily require seizures (i.e. plural).
2. Line 88: delete ‘necessarily’
3. Lines 114 and 433-435: The outcome of a coherence study can be significantly affected by the signal of the reference electrode (e.g. Schiff Neuroinformatics 2005 3:315). The authors appear to account for this issue as much as possible, but they do not mention it explicitly. Furthermore, how was the reference signal obtained? (Is it based on a single electrode or an average potential across the strip?)
4. Line 139 and Fig. 2a: Adding a trace of the signal in Fig. 2 with an indication of the pre, early, middle, and late intervals would be very helpful.
5. Paragraph starting at line 170 and the associated Fig. 3: Please specify the frequency band employed for the phase/delay calculation in this part of the manuscript. In the current version, the reader has to search through the Methods to find it.
6. Paragraph starting at line 275: The role of potassium in seizure propagation has been previously presented by others, and a reference to their work seems appropriate (e.g. Kager et al. J. Neurophysiol. 2000 84:495; Wei et al J Neurosci. 2014 34: 11733). In the description in Methods (line 547), a depolarization block is included in the model but it is not mentioned in the model description in the Result section. In addition, the depolarization block was employed in a recent study explaining seizure propagation in human cortex and it may be useful to refer to this work (Meijer et al. J Math. Neurosci. 2015 5:7).
7. Line 290: How was the spatial aspect (30 × 30 cm) included in the model? Because this aspect was unclear, I cannot relate the cartoon in Fig. 5a and the detailed spatial simulation in Fig. 5c.
8. I find the spatial plots in Fig. 5c intriguing because they seem to show the emergence of secondary focus-like spots – perhaps worthwhile to include a comment about this in the paper.

9. The paragraph starting at line 290: I think that the current description of the model dynamics is potentially confusing. Although I do agree that the model dynamics are a part of the results, the onset and offset of the network's seizure dynamics is simply the result of turning the seizure focus on and off! Therefore, the dynamics only reflect the response of the system to the (given) activity in the seizure focus. To avoid confusion, this limitation must be made very clear. The sentence starting in line 307, in particular, is easily misinterpreted. The same comment is valid for the paragraph starting at line 355 in the Discussion.
10. Line 335: This study overstates the fact that it is the first study on multi-scale seizure propagation. In fact, the relationship between propagation across a microelectrode and ECoG arrays was recently described in Smith et al. *Nat Commun.* 2016 7:11098 (See their Fig. 4).
11. The paragraph starting at line 374: Although the hypothesized behavior of the focal activity addresses an important point of ongoing discussion, I fail to see how the findings that are presented here specifically support a spatially fixed cortical source. In addition, it was previously reported that the direction of propagation may change during the seizure - both in experimental and clinical seizures (e.g. Trevelyan et al. *J. Neurosci.* 27:13513).
In this context I want to make two comments: First, there may be multiple mechanisms at work across different types of seizure or even across different seizures of a similar type. Second, limited propagation at a microelectrode scale may be considered stationary at a macroscopic scale.
12. Line 390: A depolarization block of the inhibitory neurons might sustain the seizure if gap junctions are affected by seizure activity (see Meijer et al. *J Math. Neurosci.* 2015 5:7). Since the model includes gap junctions as well as depolarization block, this statement would be simple to check and make a stronger case than making qualitative assumptions about the gap junctions alone. In the context of this discussion, I would find it extremely helpful if the traces in Fig. 5b would be extended to include plots for potassium, presence of depolarization block, and the conductivity of the gap junctions.
13. Line 391, 392: The statement about "treating the dynamics" is not clear to me.
14. Line 495, 496: I don't understand the statement about causality - relating delays to propagation of the seizure appears to be a causal assumption.
15. Line 603: Please add the Journal name to the reference.
16. Figure 2: The values of the intercepts in panels (a) and (c) do not seem to correspond with the ones plotted in (b) and (d) respectively. Please report the frequency band used for the coherence in the caption.
17. Figure 4: In panel (a), the arrows indicating propagation are hard to see. The choice of color for the graphs in panel (b) makes it difficult to identify the contributions of the different seizures.
18. Figure 6 does not add much in my opinion and, as mentioned in pt. 11, I also disagree with that part of the discussion because the claim is not supported by the data that is presented.

Wim van Drongelen
Professor of Pediatrics, Neurology, Computational Neuroscience
Technical Director Pediatric Epilepsy Center
Research Director Pediatric Epilepsy Program
Senior Fellow Computation Institute
The University of Chicago
KCBBD 4124, 900 E 57th Street, Chicago, IL

Reviewer #2 (Remarks to the Author)

This is a very interesting work studying human seizures recorded at two spatial scales.

The primary finding is that the seizures studied generate dynamics that can be accounted for by a single cortical source, not necessarily the seizure focus, from which emanate increasingly coherent waves of activity at micro and macro scales of activity. The authors propose that using a widely respected model of cortical activity (Steyn-Ross et al), that increasing ions in the extracellular space can serve during a seizure to account for the coherence.

In recent work, Smith et al in this same journal, studied single cell versus local field potential dynamics across scales. In that work, consistent with their previous studies, they demonstrated evidence that the wavefront of activity may be small, slowly propagating, and require high frequencies to locate the multiunit activity and ripples. In this present work, Martinet et al use low frequencies, study local mean field potentials rather than multiunit activity, and demonstrate that the traveling waves emanate from one cortical source. Both studies suggest that tracing the source of the waves might be a useful adjunct to focus or source localization, but one group finds that we should trace backwards, and the other group forwards! In Smith, much of the focus on explanatory effort appears at times a bit biased towards the prior experimental and theoretical postulates of that same group. In this present work, there may be a bit less self referential bias, and the introduction of the ionic fluxes to help provide a binding mechanism to explain their more macroscopic observations is attractive. The introduction of the computational model is quite compelling in offering a plausible mechanism that underlies the observations seen.

So, what is a fan of these two groups' efforts to do? We have different multi scale measurements from seizures that lead to different conclusions. The finding of propagating waves of activity are the same, but the microscale small moving wavefront is not at all consistent with the present work's macroscopic findings. The conclusions regarding seizure termination are different. The present work mapped the electrodes onto the 3D cortex, and the distances calculated should be more accurate than in Smith et al. Might this account for some of the discrepancies?

We have all been hammered on for reproducibility in science of late. Both of these studies have small numbers of patients, and the seizure dynamics can be quite different between patients (Figure 4b is dramatic in this present work). One hopes that the seizures in New York are similar enough as the seizures in Boston. But alternatively, they may be different enough samplings so that the results are not consistent with each other. This point deserves some discussion. If the seizures are dissimilar, then the results need not reflect the same dynamics. If we think that they are similar enough, then this paper will be viewed in terms of how it relates its findings to Smith et al. Both seem to contain critical aspects of seizure dynamics, and it is my opinion that both must be absorbed by the scientific community.

The one thing I am pretty sure of is that the conclusions of these two works are incommensurate as figure 6 of this present work illustrates. Perhaps the most important addition by these authors

might be to propose ways to settle this discrepancy through further experiments – literally how might you falsify, or unify, these very different findings? Would love to see that as the end of the discussion (rather than the word potassium).

Relatively minor to moderate things that distress this reviewer:

The use of the phrase ‘extracellular ions’ throughout the text I find maddening. The origin of such phraseology likely stems from the Steyn-Ross papers. Physicists. But the authors of this present work are thinking potassium when they say this, and first they pick K for a variable in equations 1 and 2, and then despite the use of generic ‘ions’ in figure legend 5a, their figure panel shows $[K^+]$, the chemical term for potassium ion concentration. What about the counter-ion chloride? Depending on extracellular volume changes, Na^+ can also vary in important ways during activity. Depending on the part of the text of this present work, a naïve reader (e.g. a physicist) might think that the authors were talking about total ion concentration, but there are no considerations of osmolarity here. My suggestion is to clean up the terminology and let the K^+ out of the closet here. If I am reading this correctly, this is a K^+ ion hypothesis that binds the experimental findings together.

That said, the theory (despite the citations) regarding gap junction function dependent upon pH changes due to K^+ is narrow. It is correlated with K^+ flux, and K^+ displaces H^+ in many acid base reactions, but there are many other sources of the large pH flux from intracellular to extracellular during seizures, including CO_2 production, lactic acid production, etc, etc. This does not alter any of the findings, but merely alters the mechanistic explanation that I find so simplistic as to have warranted this whole paragraph of gripes.

Time. The time constants in equations 2 seem way too long (line 565). They are longer than any of the data or computational windows used in this paper! I presume this is a typo. Certainly the resting transmembrane resting potential does not react to K^+ changes with a time scale of 250 seconds. And what is $\tau_{e/i}$ in line 568?

Trivia:

Line 173: it is not ‘each micro- and macro pair’ – the micro electrodes were averaged for this if I am reading the methods correctly (and line 183 correctly).

Line 253: not ‘circular distances’, but ‘circular direction differences’

Line 446: FIR but not phase preserving with, for instance, forward and backwards filtering? Does any phase distortion affect the coherency results?

Line 451: earliest or latest time?

Line 483: why 39 tapers (for the aficionados of multitaper analysis)

Line 484: which confidence level? Thomson F or bootstrap from the tapers?

Line 513: atan2 looks like computer code slipping in

Line 549: does this mean that you will enforce depolarization block below -60 mV?

Reviewer #3 (Remarks to the Author)

A. Summary of the key results

MS looks at the electric activity during focal-onset seizures in humans with temporal lobe epilepsy at two spatial levels: widespread invasive ECoG and a micro-electrode array in a single location close to the seizure onset area.

Coherence is used to calculate the phase similarity of the signals at the two spatial levels. The coherence significantly increases within the seizure. Analysis of the coherent sections points towards the evolution of large-scale propagating waves of activity during the second half of the seizure. The propagating waves appear to originate in a stable local source of abnormal activity.

A neural field model of a spatial cortical sheet with an added freely diffusing variable and locally increased excitation produces qualitatively related dynamics.

B. Originality and interest: if not novel, please give references

The topic is of eminent interest as it permits detailed insight into abnormalities of human brain activity under in vivo conditions.

The study fits in with recent interest in a more detailed understanding of the dynamics of focal-onset epileptic seizures. A large number of investigations have been made on the same data set before but not the quantitative evolution of similarity during seizures. The question of spatio-temporal patterns at different scales of observation is also the subject of reference 25 (incomplete citation).

C. Data & methodology: validity of approach, quality of data, quality of presentation

Data are of very high quality; the approach is valid; presentation is professional. Added value could lie in the combination of data analysis and the use of a mathematical model of the seizing cortical tissue but see comments below.

D. Appropriate use of statistics and treatment of uncertainties

Use of statistics is appropriate but general conclusions are necessarily limited due to the small number of cases (7 seizures from 3 patients).

E. Conclusions: robustness, validity, reliability

The strength of the conclusions is limited by the small number of sampled seizures. Using a mathematical modelling approach to complement the data analysis is potentially a way out of the dilemma.

The mathematical modelling uses a previously published homogeneous neural field equation with a local inhomogeneity to model a stable local source of instability. The wave propagation and spreading of seizure activity is then a self-organised process as long as the local source is switched on. A key modification of the original model consists in the addition of a variable representing a diffusing species and of variables to feedback the effect of that diffusing species on local cortical activity.

A problem here is that while the assumption that the diffusing species is the extracellular potassium level is a testable hypothesis, this hypothesis is not tested. It is therefore premature to assume that the qualitative similarity of the results support the mechanistic assumptions.

It has been argued previously that rather than picking a single mechanism and adjusting the model parameters until they yield results that resemble the data, testing of alternatives might be a better strategy, Wang et al PLoS Comp Biology (2014). Specifically, the model used in the MS allows for direct implementation of a null hypothesis (e.g. the source of the waves is created by random perturbations of an excitable patch of tissue) which is not done in the current MS.

F. Suggested improvements: experiments, data for possible revision

The current MS implements a specific model with a fixed local source and proposes this as an alternative to the previously published proposal of a moving source. I think this needs addressing:

- 1 The same type of data analysis as previously published needs to be done to check whether the current data sets are of the same or of a different type. If closely related, a combination of the existing data sets could be used to improve the statistics.

- 2 The mathematical model should implement mechanisms based on both alternatives and check what analysis features are best to distinguish between the two. Also, there is the possibility that both mechanisms might be found in a heterogenous patient cohort.

3 One wonders why this specific model (reference 46) was chosen. It was originally proposed to explain cortical dynamics under anesthesia, whereas numerous alternative models exist for the description of cortical seizure activity. Specifically, no comment is made on the special feature of this model, namely the vicinity of both a Hopf and a Turing instability, and a parameter region of coexistence of the two. The simulations shown in the MS seem to indicate that the model operates under similar conditions and that this model feature might be a key prerequisite of the observed dynamics (rather than the specific choice of the diffusing extracellular substance).

G. References: appropriate credit to previous work?

Reference to paper by Wang et al PLoS Comp Biology (2014) should be included.

Reference 25 should be discussed with respect to the outcomes concerning source of spreading wave activity.

H. Clarity and context: lucidity of abstract/summary, appropriateness of abstract, introduction and conclusions

MS is well-written. Title and Abstract: I am not sure I fully understand the repeated mention of "couple" or "coupling" across spatial scales. Coherence is a linear measure of amplitude similarity in the frequency domain. So the increase in coherence between scales presumably means an increase in the length of spatial order of the cortical dynamics but not necessarily a "coupling" between the spatial scales.

In conclusion, the MS is of high quality and warrants an invitation to revise before a final decision is made.

Note to Reviewers: We thank the three Reviewers for their careful consideration of our original manuscript. Please find below our responses to each Reviewer suggestion, all of which we believe have substantially improved the clarity and results of the manuscript.

Note to Editors: We have updated the manuscript to comply with the reporting requirements and format requirements of Nature Communications.

Reviewer #1 (Remarks to the Author):

The manuscript entitled “Human seizures couple across spatial scales through traveling wave dynamics” by Martinet et al., is a well written report on neocortical seizure propagation outside the epileptic focus. The experimental data provide insight into the propagation of seizure activity across two spatial scales in three patients. The presentation of the results is straightforward and easy to interpret, especially since the propagation properties at the micro- and macro-electrode scales are similar. The signal analysis is appropriate and the associated modeling approach is extremely useful and seems to be capable of explaining observations. Overall, I find the data and analyses this paper of high interest to the field, but encourage a revision addressing the points listed below.

We thank the Reviewer for the interest in this work, and careful consideration of the manuscript. Please find below our responses to each comment.

R1.Q1. Line 18: The latest definition of epilepsy (ILAE 2014) does not necessarily require seizures (i.e. plural).

R1.A1. As suggested, we have updated the text in the revised manuscript and replaced many instances of “seizures” with “seizure”.

R1.Q2. Line 88: delete ‘necessarily’

R1.A2. As suggested by the Reviewer, we have updated this line to read:

*“We therefore examine interactions between spatial scales from cortical regions recruited into seizure, and not **necessarily** interactions between the seizure focus and rest of cortex [23,28,31].”*

R1.Q3. Lines 114 and 433-435: The outcome of a coherence study can be significantly affected by the signal of the reference electrode (e.g. Schiff Neuroinformatics 2005 3:315). The authors appear to account for this issue as much as possible, but they do not

mention it explicitly. Furthermore, how was the reference signal obtained? (Is it based on a single electrode or an average potential across the strip?)

R1.A3. As suggested, we have updated the manuscript to clarify how the macroelectrode reference was chosen:

“...The reference was a strip of electrodes placed outside the dura and facing the skull at a region remote from the other grid and strip electrodes. One to four electrodes were selected from this reference strip and connected to the reference channel.”

And to clarify how the microelectrode reference was chosen:

“...The MEA consist of 96 recording platinum-tipped silicon probes, with a length of either 1-mm or 1.5-mm, corresponding to neocortical layer III as confirmed by histology after resection. The reference was an electrode placed either subdurally or epidurally and remote from the recording cite [23].”

We now mention in the revised manuscript the difficulty of interpreting the coherence from referenced data and include two new citations. We also propose that, because we observe spatial organization in the delays (consistent with a traveling wave), we do not expect the coherence results are dominated by spurious reference effects:

“... Repeating the analyses using a smaller window of 2 s and a time-bandwidth product of 4 lead to qualitatively similar results (not shown). We note that the interpretation of coherence, or any coupling measure, from referenced or re-referenced data requires care⁷³. Here we chose not to re-reference the data. Because we compute the coherence between many electrode pairs across spatial scales with a different physical reference at each spatial scale, and then analyze the spatial organization of the delays inferred from this coherence, we would expect to find no evidence of traveling waves if the coherence results were dominated by spurious effects⁷⁴. However, we do find evidence for traveling waves, which suggests that these results are not dominated by a reference signal. We also note that the model data, which is reference-free, produces dynamics consistent with the in vivo data; this consistency supports a relatively quiet in vivo reference.”

R1.Q4. Line 139 and Fig. 2a: Adding a trace of the signal in Fig. 2 with an indication of the pre, early, middle, and late intervals would be very helpful.

R1.A4. As recommended by the Reviewer, we now indicate these intervals in Figure 1b:

And, we have updated the caption of Figure 1 to read:

“(b,c) Example voltage traces recorded simultaneously from macroelectrodes (upper) and microelectrodes (lower) during (b) seizure, and during (c) a single spike-and-wave event. The green vertical bars indicates the same time point in both subfigures. The blue vertical bars and labels in (c) correspond to the voltage maps in (d). Four intervals (pre-seizure, early seizure, middle seizure, and late seizure) are indicated in (b). Scale bar in (c) indicates 100 ms.”

We also now include an explicit reference to Figure 1b in the caption of Figure 2 when describing these intervals:

“(a) Example average inter-scale coherence between the microelectrodes and macroelectrodes versus distance for a single patient and seizure. Each dot represents the coherence and distance of a macroelectrode during four intervals; grey, pre-seizure; pink, early seizure; red, middle seizure; maroon, late seizure (Figure 1b). The lines indicate linear regression estimates for each interval.”

R1.Q5. Paragraph starting at line 170 and the associated Fig. 3: Please specify the frequency band employed for the phase/delay calculation in this part of the manuscript. In the current version, the reader has to search through the Methods to find it.

R1.A5. As recommended, we have updated the manuscript to read:

“To further characterize the spatial organization of this coupling, we use the coherence results *between 1-13 Hz* to estimate the delay between each microelectrode pair, and each micro- and macroelectrode pair (see Methods).”

R1.Q6. Paragraph starting at line 275: The role of potassium in seizure propagation has been previously presented by others, and a reference to their work seems appropriate (e.g. Kager et al. J. Neurophysiol. 2000 84:495; Wei et al J Neurosci. 2014 34: 11733). In the description in Methods (line 547), a depolarization block is included in the model but it is not mentioned in the model description in the Result section. In addition, the depolarization block was employed in a recent study explaining seizure propagation in human cortex and it may be useful to refer to this work (Meijer et al. J Math. Neurosci. 2015 5:7).

R1.A6. As suggested, we have updated the Results section of the manuscript to include additional references to the role of potassium in seizure propagation, as follows:

“We update this model to simulate the temporal evolution of seizure by including a slowly evolving variable representing the changing concentration of extracellular *potassium, which increases dramatically during seizure and other dysfunctional brain states*⁵²⁻⁵⁷.”

We have also updated the Discussion to reference the modeling work of Kager et al:

“Updating the model to address these limitations and incorporate single neuron activity with changes in extracellular ion dynamics^{56,57} would provide additional insight.”

The Reviewer is correct that we mention the model of depolarization block only in the Methods. We do so because, in our model formulation, depolarization block does not play an important role. We now state this observation in Methods, and include Supplementary Figure 4 to illustrate

this point. Please see R1.A12 below for more details (which includes a new reference to [Meijer et al 2015] as well).

R1.Q7. Line 290: How was the spatial aspect (30 × 30 cm) included in the model?

Because this aspect was unclear, I cannot relate the cartoon in Fig. 5a and the detailed spatial simulation in Fig. 5c.

R1.A7. We apologize that this was unclear. In the simulation, we divided a 30 cm by 30 cm plane into 100 by 100 discrete elements (i.e., a spatial resolution of 0.3 cm). Therefore, each element in the model corresponds to a 3 mm by 3 mm area of cortex. To make this more clear, we have updated Figure 5a and Figure 5c to include scale bars:

Both scale bars are now defined in the Figure 5 caption. In (a), the scale bar indicates 3 mm. In (c), the scale bar indicates 10 cm.

R1.Q8. I find the spatial plots in Fig. 5c intriguing because they seem to show the emergence of secondary focus-like spots – perhaps worthwhile to include a comment about this in the paper.

R1.A8. This is an interesting observation. In this model, we use the term “focus” to indicate the region of cortex with increased cell resting potential (we note that, of course, this is a simplification to reflect the increased excitability of the focus). In the simulation, other brain areas become activated (i.e., develop higher activity) after initiation of the focus; however, these subsequently activated areas do not possess an increased cell resting potential. We note that reducing the cell resting potential of the focus after the spot pattern has developed (i.e., reducing the cell resting potential of the focus after frame (i) in Figure 5c), we find that the spot pattern evolves to local waves (as in frame (ii) in Figure 5c), and then into more global waves

that propagate across the entire simulated domain. However, unlike the results presented in Figure 5c, the traveling waves in this case do not emerge from a fixed spatial location. Instead, the spatiotemporal pattern remains much more complex (see Reviewer Figure 1 immediately below this text). We also find that, in this scenario, the seizure does not terminate in the same way. In the original simulation, when we inactivate the focal source late in seizure, the seizure activity stops. This same termination scenario does not work in Reviewer Figure 1, because there is no focal source to inactivate. Instead, a global change is required to end the seizure, such as reducing the excitation of the entire cortical sheet.

Reviewer Figure 1. In this simulation, we remove the cortical source of increased cell resting potential after the emergence of spots (after the 6th frame above). Like the simulations in Figure 5c, the spatiotemporal pattern still evolves from spots, to local propagation, to more global propagation. However, the more global propagation (e.g., the third row of figure) lacks a spatially fixed source that emits the traveling waves.

We now comment on this observation in the revised manuscript as follows:

“Reducing the source excitability eliminates the wave source and the cortical sheet returns to quiescence (Figure 5c, label iv). We note that reducing the source excitability earlier in the seizure, after the emergence of the spatial patterns in Figure 5, label i, produces a similar sequence of spatiotemporal states. However, in this case, the traveling waves that appear late in the seizure lack a fixed source, and an approximately simultaneous seizure termination across space – as observed in the human recordings – would require a more global intervention, such as reducing the excitability of the entire cortical sheet.”

R1.Q9. The paragraph starting at line 290: I think that the current description of the model dynamics is potentially confusing. Although I do agree that the model dynamics are a part of the results, the onset and offset of the network’s seizure dynamics is simply the result of turning the seizure focus on and off! Therefore, the dynamics only reflect the response of the system to the (given) activity in the seizure focus. To avoid confusion, this limitation must be made very clear. The sentence starting in line 307, in particular, is easily misinterpreted. The same comment is valid for the paragraph starting at line 355 in the Discussion.

R1.A9. In retrospect, we agree that our original description of this component of the model dynamics was confusing. To correct this, we have clarified the updated manuscript as follows:

~~“We note that these model dynamics evolve with no externally applied parameter manipulations; instead, In this model, the slow evolution of the extracellular ion concentration navigates the dynamics between the different spatiotemporal stages.”~~

We have also updated the identified paragraph in the Discussion to read:

“This dynamical understanding is consistent with a computational model in which traveling waves emerge from a cortical source. To simulate a seizure in the model, we increase the activity of the cortical source. Initially, this source is difficult to identify, as a complex pattern of activation and inactivation appears over the simulated cortical surface. For this cortical source to emerge spontaneously requires the slow accumulation and diffusion of extracellular potassium in the model.”

R1.Q10. Line 335: This study overstates the fact that it is the first study on multi-scale seizure propagation. In fact, the relationship between propagation across a microelectrode and ECoG arrays was recently described in Smith et al. Nat Commun. 2016 7:11098 (See their Fig. 4).

R1.A10. We agree that, in retrospect, this claim was too strong. We have updated this paragraph in the Discussion to state more clearly that this is the first study to compute the coherence between the two spatial scales:

“The seizing human cortex provides an ideal system in which to study the spatiotemporal dynamics of multi-scale brain activity. A seizure is a stereotyped multi-scale dysfunction proposed to initiate at a (microscopic) source and – in these patients – subsequently recruit all or large portions of the entire observable cortex. To understand the multi-scale, spatiotemporal dynamics of seizure requires assessment of the temporal and spatial organization of simultaneous macroscopic and microscopic voltage recordings from human cortex. These invasive observations, which involve implantation of a subdural clinical macroelectrode array and high-density microelectrode array, are only performed in human patients with refractory epilepsy. Other studies have investigated features of macroelectrode and microelectrode data recorded simultaneously^{25,31,61–63}, but have not directly assessed the coherence between these two spatial scales.”

R1.Q11. The paragraph starting at line 374: Although the hypothesized behavior of the focal activity addresses an important point of ongoing discussion, I fail to see how the findings that are presented here specifically support a spatially fixed cortical source. In addition, it was previously reported that the direction of propagation may change during the seizure - both in experimental and clinical seizures (e.g. Trevelyan et al. J. Neurosci. 27:13513). In this context I want to make two comments: First, there may be multiple

mechanisms at work across different types of seizure or even across different seizures of a similar type. Second, limited propagation at a microelectrode scale may be considered stationary at a macroscopic scale.

R1.A11. In the revised manuscript, we attempted to provide additional support for the hypothesis of a spatially fixed cortical source by adding new model results for a simulated ictal wavefront (please see the new Figure 6, and the associated text in Results in R2.A5). We have also updated the Discussion to include a more detailed comparison of the results in [Smith et al., Nat Comm 2016] and the work presented here. As suggested, in this revised Discussion, we note that both scenarios may occur in a heterogeneous patient cohort. Please see R2.A5 for the new Discussion text.

We agree that limited propagation (or spatially disorganized activity) at the microelectrode scale may - as this disorganized activity aggregates - appear as stationary activity at the macroelectrode scale. Here we did not focus on this issue, and instead focused on the emergence of highly organized activity (traveling wave propagation) across spatial scales. In the future, understanding how more local organization within the microelectrodes manifests at the macroelectrode scale would be a very interesting topic.

R1.Q12. Line 390: A depolarization block of the inhibitory neurons might sustain the seizure if gap junctions are affected by seizure activity (see Meijer et al. J Math. Neurosci. 2015 5:7). Since the model includes gap junctions as well as depolarization block, this statement would be simple to check and make a stronger case than making qualitative assumptions about the gap junctions alone. In the context of this discussion, I would find it extremely helpful if the traces in Fig. 5b would be extended to include plots for potassium, presence of depolarization block, and the conductivity of the gap junctions.

R1.A12. As recommended, we now include in the revised Supplementary Material three new figures.

Supplementary Figure 4. As the Reviewer may have guessed, we included in the model depolarization block, motivated by the results in [Meijer et al., J Math Neurosci 2015], and discussions with the first and last author of that work! However, we find that - in the model presented here - depolarization block does not impact the simulated dynamics. To illustrate this, we plot in Supplementary Figure 4 the firing rate versus voltage curve used in our model, which has an approximate Gaussian shape (as in [Meijer et al., J Math Neurosci 2015]). In addition, we show in this figure the voltage and firing rate values obtained in a simulated seizure (i.e., a seizure simulated as in Figure 5b,c). We observe that neither the excitatory nor inhibitory populations enter depolarization block; in both populations, the voltage values do not depolarize enough to enter the region of depolarization block in the model. Despite this, we decided to

keep depolarization block in the model. This result was not made clear in the original manuscript. We now make this result clear in the revised manuscript as follows:

“... (ii) We included a model of depolarization block, so that the firing rate of each population approaches zero as the voltage exceeds -20 mV; i.e., we apply a Gaussian activation function, rather than the standard sigmoid function⁷⁸. Some recent observations from human seizing cortex suggest an important role for depolarization block^{78,79}, while others suggest this role may depend on the type of seizure^{23,80}. We do not find a critical role for depolarization block in the large amplitude, low frequency dynamics simulated here (Supplementary Figure 4). However, to simulate the low amplitude, high frequency rhythms commonly observed at seizure onset would require inclusion of additional mechanisms, for example an additional cell population, in which depolarization block may serve an important role.”

Supplementary Figure 1. As suggested by the Reviewer, we now include the new Supplementary Figure 1 to illustrate the dynamics of potassium and the conductivity of the gap junctions during a simulated seizure. We refer to this Supplementary Figure in the main text as follows:

“As time evolves, the extracellular potassium concentration increases in the active regions, thereby increasing the excitability and reducing the gap junction strength in these regions (Supplementary Figure 1).”

Supplementary Figure 3. Finally, as suggested by the Reviewer, we have strengthened the qualitative statements about potential therapies by now including the new Supplementary Figure 3, which shows in simulation the impact of preventing potassium accumulation or preventing the loss of inhibitory gap junctions. With these changes, the transition to traveling waves does not occur. Instead, the dynamics remain in a fixed spatial pattern. We have updated the text to read:

“Motivated by the computational model, potential therapies could target the accumulation of extracellular potassium, for example by increasing glial uptake or developing a physical collection mechanism, or act to preserve inhibitory gap junctions. Both modifications prevent the emergence of traveling waves in the model (Supplementary Figure 3).”

R1.Q13. Line 391, 392: The statement about “treating the dynamics” is not clear to me.

R1.A13. By “treating the dynamics” we meant preventing the propagation of the traveling waves of seizure. We have revised this text to make our meaning clearer, as follows:

“Second, as described above, we propose targeting the source of traveling waves that emerge during seizure. Motivated by the in vivo data and computational model, potential therapies could target the cortical source - rather than the seizure source, which may be inaccessible; firewall the cortical source⁷⁰; or disrupt wave propagation in some way to prevent waves from reaching eloquent cortex.”

R1.Q14. Line 495, 496: I don't understand the statement about causality - relating delays to propagation of the seizure appears to be a causal assumption.

R1.A14. We now omit this sentence from the revised manuscript:

“Otherwise, the delay between the electrode pair was considered undefined. ~~We note that the existence of a delay between two electrodes is consistent with a causal relationship, but we did not interpret these associations as causal here.~~”

R1.Q15. Line 603: Please add the Journal name to the reference.

R1.A15. Thank you for identifying this omission. We have updated this reference.

R1.Q16. Figure 2: The values of the intercepts in panels (a) and (c) do not seem to correspond with the ones plotted in (b) and (d) respectively. Please report the frequency band used for the coherence in the caption.

R1.A16. To help illustrate the correspondence, we have included Reviewer Figure 2. In this figure, we show that the values of the intercepts in panels (a) and (c) of Figure 2 correspond to a single seizure and thus are represented by a single point for each time interval in panels (b) and (d), shown here as points with red circles:

As suggested, we have updated the caption of Figure 2 to include the frequency band used to compute the average coherence:

“(a) Example average inter-scale coherence (1-13 Hz) between the microelectrodes and macroelectrodes versus distance for a single patient and seizure.”

R1.Q17. Figure 4: In panel (a), the arrows indicating propagation are hard to see. The choice of color for the graphs in panel (b) makes it difficult to identify the contributions of the different seizures.

R1.A17. In the revised manuscript, we have updated Figure 4 to address both issues. We include the revised figure here:

R1.Q18. Figure 6 does not add much in my opinion and, as mentioned in pt. 11, I also disagree with that part of the discussion because the claim is not supported by the data that is presented.

R1.A18. To further support the claim that a fixed cortical source produces activity consistent with the *in vivo* data, while an ictal wavefront does not appear to be consistent with the data analyzed here, we have updated Figure 6 to include a new simulation result of the ictal wavefront. We discuss this new result in the revised manuscript; please see R2.A5 for details. We hope that, by including this simulation, the simple cartoon illustration (Figure 6c,d in the revised manuscript) helps provide intuition for the main differences between the fixed cortical source versus ictal wavefront scenarios.

Reviewer #2 (Remarks to the Author):

R2.Q1. This is a very interesting work studying human seizures recorded at two spatial scales. The primary finding is that the seizures studied generate dynamics that can be

accounted for by a single cortical source, not necessarily the seizure focus, from which emanate increasingly coherent waves of activity at micro and macro scales of activity. The authors propose that using a widely respected model of cortical activity (Steyn-Ross et al), that increasing ions in the extracellular space can serve during a seizure to account for the coherence.

R2.A1. We are glad the Reviewer found this work interesting.

R2.Q2. In recent work, Smith et al in this same journal, studied single cell versus local field potential dynamics across scales. In that work, consistent with their previous studies, they demonstrated evidence that the wavefront of activity may be small, slowly propagating, and require high frequencies to locate the multiunit activity and ripples. In this present work, Martinet et al use low frequencies, study local mean field potentials rather than multiunit activity, and demonstrate that the traveling waves emanate from one cortical source. Both studies suggest that tracing the source of the waves might be a useful adjunct to focus or source localization, but one group finds that we should trace backwards, and the other group forwards! In Smith, much of the focus on explanatory effort appears at times a bit biased towards the prior experimental and theoretical postulates of that same group. In this present work, there may be a bit less self referential bias, and the introduction of the ionic fluxes to help provide a binding mechanism to explain their more macroscopic observations is attractive. The introduction of the computational model is quite compelling in offering a plausible mechanism that underlies the observations seen.

R2.A2. Again, we are glad the Reviewer appreciates the inclusion of the computational model. As suggested by the Reviewer, we have updated the Discussion in the revised manuscript to include a more detailed comparison of the results presented here and in [Smith et al, Nat Comm, 2016]. Please see R2.A5 below for more details.

R2.Q3. So, what is a fan of these two groups' efforts to do? We have different multi scale measurements from seizures that lead to different conclusions. The finding of propagating waves of activity are the same, but the microscale small moving wavefront is not at all consistent with the present work's macroscopic findings. The conclusions regarding seizure termination are different. The present work mapped the electrodes onto the 3D cortex, and the distances calculated should be more accurate than in Smith et al. Might this account for some of the discrepancies?

R2.A3. As suggested by the Reviewer, we attempt in the revised manuscript to make clear these important differences between the results reported here and in [Smith et al., Nat Comm, 2016]. Please see R2.A5 below for more details.

As recommended by the Reviewer, we have repeated our analysis using the “electrode distances” rather than the physical distances determined from the 3D cortex. We find the results are qualitatively similar, as shown in the figure provided here for the Reviewer (compare to Figure 2 of the manuscript):

We have updated the manuscript to mention the qualitatively similar results obtained using the electrode distance, as follows:

“To compute the distance between a macroelectrode and the microelectrode array, we used the co-registered 3D positions of the macroelectrode on the brain surface (see 74 and section Anatomical Figures) to estimate the geodesic distance. Repeating the analysis of coherence versus distance (Figure 2) using the macroelectrode grid spacing of 1 cm, we find qualitatively similar results (not shown).”

R2.Q4. We have all been hammered on for reproducibility in science of late. Both of these studies have small numbers of patients, and the seizure dynamics can be quite different

between patients (Figure 4b is dramatic in this present work). One hopes that the seizures in New York are similar enough as the seizures in Boston. But alternatively, they may be different enough samplings so that the results are not consistent with each other. This point deserves some discussion. If the seizures are dissimilar, then the results need not reflect the same dynamics. If we think that they are similar enough, then this paper will be viewed in terms of how it relates its findings to Smith et al. Both seem to contain critical aspects of seizure dynamics, and it is my opinion that both must be absorbed by the scientific community.

R2.A4. We agree that this is an essential point. We now mention this important observation - and additional comparisons - between the results in the two studies in the revised Discussion. Please see R2.A5 below for more details.

R2.Q5. The one thing I am pretty sure of is that the conclusions of these two works are incommensurate as figure 6 of this present work illustrates. Perhaps the most important addition by these authors might be to propose ways to settle this discrepancy through further experiments – literally how might you falsify, or unify, these very different findings? Would love to see that as the end of the discussion (rather than the word potassium).

R2.A5. We agree with the Reviewer's objectives in this question. We were somewhat reluctant to tackle these discrepancies head on. But, the Reviewer, quite appropriately, has identified an important – and sensitive – topic for discussion. Consequently, we have updated the manuscript in two ways. First, we now include in the last two paragraphs of Results a new simulation which contrasts the main features of these two works (an ictal wavefront as proposed in [Smith et al., 2016] and a fixed cortical source as proposed here) and describes how these conclusions differ (revised Figure 6):

“...The direction consistency increases during seizure at both spatial scales ($p < 5e-3$, t-test, Figure 5e), while the velocities simulated are consistent with the in vivo values; both range between 50-400 mm/s (Figure 5f). We note that, if instead the source of increased excitability appears at random spatial locations over time, then the direction consistency during seizure is significantly smaller ($p < 0.005$, t-test) in this model (mean 0.47, standard deviation 0.22, during the late seizure interval) compared to the in vivo data (mean 0.81, standard deviation 0.16 during the late seizure interval; Supplementary Figure 2 and Figure 6b). Finally, the source directions align during seizure (Figure 5g), consistent with traveling waves that propagate in the same direction across the micro- and macroelectrodes. These results support the conclusion that an established mean-field model, updated to mimic changes in extracellular potassium dynamics, simulates important features consistent with the in vivo seizure data.

A related scenario of seizure evolution has been proposed^{25,26}. In the most basic formulation of that scenario, the seizing territory expands as a slowly advancing, sharply demarcated, narrow (< 2 mm) band

of multiunit firing, termed the ictal wavefront. Traveling waves arise behind the ictal wavefront as it slowly and radially expands across the cortex. These traveling waves, which consist of fast-moving synaptic potentials, produce the low-frequency, large amplitude electroencephalogram (EEG) signature of seizures over broad areas of cortex. This scenario benefits from both clinical^{25,26} and experimental⁶⁰ observations.

Our model provides a framework to simulate and compare the ictal wavefront scenario^{25,26} with the scenario of a fixed cortical source as proposed here. To do so, we consider the simplest formulation of an ictal wavefront: a two-dimensional boundary of increased excitation that spreads outward at an approximate speed of 1 mm/s²⁶. This spatial spread includes a random component, so that the ictal wavefront appears as a distorted circle in the two-dimensional plane (Supplementary Methods). This distortion reflects the fact that the ictal wavefront is unlikely to spread as a perfect radial wave across different cortical areas, gyri and sulci. Behind this wavefront, in the seizing territory, traveling waves emerge consistent with the classic EEG signature of seizures (Figure 6a). Computing the direction consistency approaching seizure termination for the microelectrode data simulated in this model, we find significantly smaller values (mean 0.40, standard deviation 0.11) compared to the in vivo data (mean 0.81, standard deviation 0.16; $p < 1 \times 10^{-4}$, t-test) and compared to the model with a fixed cortical source (mean 0.89, standard deviation 0.1; $p < 1 \times 10^{-8}$, t-test; Figure 6b). In this simulation, in which the entire ictal wavefront remains active, different locations on the expanding ictal wavefront emit traveling waves, which then propagate differently across the microelectrode array over time, thus reducing the direction consistency (Figure 6c). We do not find a significant difference ($p=0.22$, t-test) between the direction consistency computed for the in vivo data and the model with a fixed cortical source. In this scenario, the fixed spatial location of the cortical source results in waves that travel consistently across the microelectrode array (Figure 6d). However, we note that, if only a small region of the ictal wavefront remained active before seizure termination, then this region could also act as a slowly drifting source of cortical waves.”

Figure 6: Simulations of an ictal wavefront produce direction consistency measures inconsistent with the in vivo data.

(a) Example spatial maps of simulated activity for the simplest ictal wavefront scenario. The arrangement and color scale are the same as in Figure 5a. An ictal wavefront emerges (i) and slowly recruits cortical territory. As the ictal wavefront expands, traveling waves propagate into the recruited territory from different directions; compare (ii) and (iii).

(b) The direction consistency during the last half of seizure in three simulation scenarios and for the in vivo data. Compared to the in vivo data, the direction consistency is significantly lower during the second half of seizure for the random source locations and ictal wavefront simulations; ** indicates $p < 0.005$, t-test.

(c,d) Schematic representations for two related scenarios of cortical wave activity during seizure. In (c), the ictal wavefront (orange) evolves in space to produce traveling waves (purple) that propagate to the microelectrode array (red) from different directions. In (d), a cortical source (orange) produces waves (purple) that impact the microelectrode array (red) from the same direction.

We have also updated the Discussion to more clearly define the similarities and differences between these works, and propose future experiments that may help clarify each group's findings:

“Numerous similarities exist between the results presented in ^{25,26} and those presented here. In both cases, traveling waves are observed with similar speeds that propagate in preferred directions consistently across the macroelectrode and microelectrode domains. These similarities occur despite the

small number of subjects analyzed (three in ²⁵ and three here, which limits general conclusions) and the different data analysis approaches employed (e.g., characterization of ictal discharges in ²⁵ and the entire field time series here). In addition, conceptual similarities link the scenario proposed in ^{25,26} and the one proposed here. Both scenarios suggest that a small cortical source projects traveling waves over a broad cortical area, that these traveling waves induce synchronization in the low-frequency field activity, and that a sufficient dissipation of the cortical source causes seizure termination. However, the two scenarios suggest a different source of ictal activity: in ^{25,26} the ictal wavefront is proposed as the source, while here we hypothesize that a fixed cortical location is the source. Using simulations that capture the basic features of these two scenarios (Figure 5 and Figure 6), we find that the fixed cortical source produces propagating waves with a direction consistency similar to the human data analyzed in this study, while a uniformly active ictal wavefront produced a significantly smaller direction consistency.

In the future, two procedures may help further distinguish these two proposed scenarios. First, in the fixed cortical source model, we hypothesize that stimulation delivered to a single cortical location - the cortical source - late in seizure will disrupt traveling wave propagation. Alternatively, we expect that single-site stimulation would not disrupt traveling waves that propagate from a uniformly active ictal wavefront; in this scenario, stimulation would disrupt only a part of the ictal wavefront, while the rest of the ictal wavefront would continue to emit traveling waves into the recruited brain region. Second, in the cortical source model, traveling waves propagate outward from the cortical source, and these waves become more salient approaching seizure termination. Therefore, we hypothesize that identifying the cortical source of these traveling waves - even late in seizure - isolates a potential treatment target, either the cortical area itself or the subcortical areas that drive it. In the ictal wavefront model, the traveling waves are less informative for identifying a treatment target; these traveling waves emerge from the ictal wavefront, which by the end of seizure has propagated away from its point of emergence on the cortex. Instead, tracking the slow evolution of the ictal wavefront may identify a candidate focal treatment target. We note that these two procedures, which exceed the immediate scope of the current study, may reveal that both scenarios occur in a heterogeneous patient cohort⁶⁹.

We conclude that the two scenarios, both of which are compatible with many aspects of the observed seizure activity, possess particular distinguishing features in terms of their mechanisms, dynamics, and response to stimulation⁶⁹. The two scenarios also differ in the mechanisms of seizure termination. To end seizure abruptly across a wide cortical region, some mechanism must weaken simultaneously the entire spatially distributed ictal wavefront. For example, the entire boundary of the ictal wavefront may encroach on a surrounding area with superior inhibitory restraint. Without this simultaneous cessation, some regions of the ictal wavefront would continue to broadcast traveling waves into recruited cortex. We note that a non-uniform collapse of the ictal wavefront could increase the consistency of traveling waves before seizure termination; for example, if only a small region of the ictal wavefront remained active, then only this region would broadcast cortical waves, which would propagate from a single direction over the brain. Alternatively, in the fixed cortical source model, mechanisms that operate over a limited cortical region could inactivate the cortical source and terminate seizure.”

Relatively minor to moderate things that distress this reviewer:

R2.Q6. The use of the phrase ‘extracellular ions’ throughout the text I find maddening. The origin of such phraseology likely stems from the Steyn-Ross papers. Physicists. But the authors of this present work are thinking potassium when they say this, and first they pick K for a variable in equations 1 and 2, and then despite the use of generic ‘ions’ in figure legend 5a, their figure panel shows [K+], the chemical term for potassium ion concentration. What about the counter-ion chloride? Depending on extracellular volume changes, Na⁺ can also vary in important ways during activity. Depending on the part of the text of this present work, a naïve reader (e.g. a physicist) might think that the authors were talking about total ion concentration, but there are no considerations of osmolarity here. My suggestion is to clean up the terminology and let the K⁺ out of the closet here. If I am reading this correctly, this is a K⁺ ion hypothesis that binds the experimental findings together.

R2.A6. We certainly do not want to distress the reviewer! As recommended by the Reviewer, we have removed the phrase “extracellular ions” and replaced it with “extracellular potassium” throughout. For example, we have updated the introduction to the model in Results, as follows:

*“We update this model to simulate the temporal evolution of seizure by including a slowly evolving variable representing the changing concentration of extracellular **potassium**, which increases dramatically during seizure and other dysfunctional brain states^{52–57}. In the model, activity of either cell population increases the local extracellular **potassium** concentration, which gradually decays (e.g., due to uptake by glial cells) and also diffuses in space. We assume that changes in the local extracellular **potassium** concentration impact the neuronal dynamics in two ways. First, we assume that an increase in the local extracellular **potassium** concentration increases the excitability of the local neural populations by increasing the reversal potential for potassium. We model this impact by increasing the resting potential of both cell populations with increasing local extracellular **potassium** concentration. Second, we assume that increases in the local extracellular **potassium** concentration act to decrease the inhibitory-to-inhibitory gap junction diffusive-coupling strength.”*

We have similarly updated other sections of the manuscript text.

R2.Q7. That said, the theory (despite the citations) regarding gap junction function dependent upon pH changes due to K⁺ is narrow. It is correlated with K⁺ flux, and K⁺ displaces H⁺ in many acid base reactions, but there are many other sources of the large pH flux from intracellular to extracellular during seizures, including CO₂ production, lactic acid production, etc, etc. This does not to alter any of the findings, but merely alters the mechanistic explanation that I find so simplistic as to have warranted this whole paragraph of gripes.

R2.A7. As suggested, we have updated this simplistic statement to indicate that other sources contribute to changes in pH during seizure:

“Second, *we assume that* increases in the local extracellular *potassium* concentration act to decrease the *inhibitory-to-inhibitory gap junction diffusive-coupling strength*. This effect is included to mimic the closing of gap junctions caused by the slow acidification of the extracellular environment *late in seizure* associated with increased extracellular potassium [60,61] *and other sources, such as the accumulation of lactic acid and CO₂*⁵³.

R2.Q8. Time. The time constants in equations 2 seem way too long (line 565). They are longer than any of the data or computational windows used in this paper! I presume this is a typo. Certainly the resting transmembrane resting potential does not react to K+ changes with a time scale of 250 seconds. And what is tau e/i in line 568?

R2.A8. In retrospect, we realize that equation 2 was confusing. In the original equations, we included two parameters (k and tau), one in the numerator and the other in the denominator, and separately adjusted each one. However, in this phenomenological model, these two parameters can be combined. Therefore, in the revised manuscript, we have removed the constants k_{ii} and k_V from equation (2). The revised manuscript now reads:

$$\begin{aligned} \frac{dD_{ii}}{dt} &= -\frac{D_{ii}}{\tau_D} \\ \frac{d\Delta V_b^{rest}}{dt} &= \frac{\Delta V_b^{rest}}{\tau_V} \end{aligned} \quad (2)$$

where $\tau_D = 4$ s and $\tau_V = 25$ s. The time constants in equation 2 are now within the computational windows (order 100 s) used in the simulations.

Trivia:

Line 173: it is not ‘each micro- and macro pair’ – the micro electrodes were averaged for this if I am reading the methods correctly (and line 183 correctly).

The statement the Reviewer identified here reads:

“To further characterize the spatial organization of this coupling, we use the coherence results to estimate the delay between each microelectrode pair, and each micro- and macroelectrode pair (see Methods).”

In this first step of the analysis, we in fact did compute the delay between each micro- and macroelectrode pair. In a subsequent step (third sentence in the paragraph following the sentence quoted above), we then averaged these delays from all microelectrode to each macroelectrode.

Line 253: not ‘circular distances’, but ‘circular direction differences’

Correct. As recommend, we have updated the text to read:

*“To characterize the differences in source direction between the two spatial scales, we compute their **circular direction difference** at each moment in time during seizure. The **circular direction differences** for each patient concentrate near 0 radians (Figure 4c).”*

We have also updated the caption in Figure 4C to read:

*“(c) The distribution of **circular direction differences** between the source directions at the microscale and macroscale during seizure concentrate near 0.”*

Line 446: FIR but not phase preserving with, for instance, forward and backwards filtering? Does any phase distortion affect the coherency results?

We apologize this was omitted. We have updated the text to read:

*“Signals from the MEA were acquired continuously at 30 kHz per channel and then subsequently down-sampled to 500 Hz by low-pass filtering (**zero-phase forward and reverse** finite impulse response filter of order 1000 with cut-off frequency of 250 Hz) and interpolating voltages at the same time-points as the ECoG.”*

Line 451: earliest or latest time?

We have updated this statement to read:

*“Seizure onset times were determined by an experienced encephalographer (S.S.C.) through inspection of the macroelectrode recordings, referral to the clinical report, and clinical manifestations recorded on video. **The seizure end time was defined as the latest time at which both the microelectrode and macroelectrode recordings displayed large amplitude ictal activity.**”*

Line 483: why 39 tapers (for the aficionados of multitaper analysis)

To address this, we have updated the text to read:

*“We divided the data into 10 s windows with 9 s overlap, beginning 60 s before seizure onset and ending at seizure termination, and computed the coherence within each window using a time-bandwidth product of 20 (bandwidth of 2 Hz) and 39 tapers, **which is chosen to be one less than the Shannon number (or one less than twice the time-bandwidth product)**⁷².”*

Line 484: which confidence level? Thomson F or bootstrap from the tapers?

To clarify this, we have updated the text to read:

*“We declared significant the values of the coherence larger than the **theoretical** confidence level at 99.5%.”*

Line 513: atan2 looks like computer code slipping in

To clarify this, we have updated the text to read:

*“... the direction of the wave source, estimated as **the four-quadrant inverse tangent, with horizontal coordinate b_1 and vertical coordinate b_2.**”*

Line 549: does this mean that you will enforce depolarization block below -60 mV?

As part of the original formulation of the Steyn-Ross model, the firing rate becomes small (less than 10 Hz) when the voltage is too low (below approximately -60 mV). We have added depolarization block to the model, so that the firing rate also becomes small (less than 10 Hz) when the voltage becomes too large (above approximately -20 mV). To make clear our addition to the model, we have updated the text to read:

*“(ii) We included a model of depolarization block, so that the firing rate of each population **approaches zero as the voltage exceeds -20 mV; i.e., we apply a Gaussian activation function, rather than the standard sigmoid function⁷⁸.**”*

Reviewer #3 (Remarks to the Author):

A. Summary of the key results

MS looks at the electric activity during focal-onset seizures in humans with temporal lobe epilepsy at two spatial levels: widespread invasive ECoG and a micro-electrode array in a single location close to the seizure onset area.

Coherence is used to calculate the phase similarity of the signals at the two spatial levels. The coherence significantly increases within the seizure. Analysis of the coherent sections points towards the evolution of large-scale propagating waves of activity during the second half of the seizure. The propagating waves appear to originate in a stable local source of abnormal activity.

A neural field model of a spatial cortical sheet with an added freely diffusing variable and locally increased excitation produces qualitatively related dynamics.

B. Originality and interest: if not novel, please give references

R3.Q1. The topic is of eminent interest as it permits detailed insight into abnormalities of human brain activity under in vivo conditions.

R3.A1. We are glad the Reviewer found this work of interest.

R3.Q2. The study fits in with recent interest in a more detailed understanding of the dynamics of focal-onset epileptic seizures. A large number of investigations have been made on the same data set before but not the quantitative evolution of similarity during seizures. The question of spatio-temporal patterns at different scales of observation is also the subject of reference 25 (incomplete citation).

R3.A2. We note that, in the revised manuscript, we now describe in more detail the similarities and differences between this work and the results in reference 25; please see R2.A5 for more details. We have also corrected the citation to reference 25.

C. Data & methodology: validity of approach, quality of data, quality of presentation

R3.Q3. Data are of very high quality; the approach is valid; presentation is professional. Added value could lie in the combination of data analysis and the use of a mathematical model of the seizing cortical tissue but see comments below.

R3.A3. We are glad the Reviewer found the data quality, approach and presentation sufficient. As suggested by the Reviewer, we further combine the data analysis and mathematical model in the revised manuscript, as described in more detail below.

D. Appropriate use of statistics and treatment of uncertainties

R3.Q4. Use of statistics is appropriate but general conclusions are necessarily limited due to the small number of cases (7 seizures from 3 patients).

R3.A4. We agree with the Reviewer, and now highlight the small number of subjects analyzed here and in the related study of [Smith et al., Nat Comm 2016] (from R2.A5):

“Numerous similarities exist between the results presented in ^{25,26} and those presented here. In both cases, traveling waves are observed with similar speeds that propagate in preferred directions consistently across the macroelectrode and microelectrode domains. These similarities occur despite the small number of subjects analyzed (three in ²⁵ and three here, which limits general conclusions) and the different data analysis approaches employed (e.g., characterization of ictal discharges in ²⁵ and the entire field time series here).”

E. Conclusions: robustness, validity, reliability

R3.Q5. The strength of the conclusions is limited by the small number of sampled seizures. Using a mathematical modelling approach to complement the data analysis is potentially a way out of the dilemma.

R3.A5. As noted by the Reviewer, we now mention this limitation in the revised manuscript. Please see our previous response (R3.A4) for the new text. We agree with the Reviewer that the model results complement the *in vivo* results. As suggested by the Reviewer, we include in the revised manuscript new model simulations that provide further support for the proposed mechanisms. Please see below (R3.A7) for more details.

R3.Q6. The mathematical modelling uses a previously published homogeneous neural field equation with a local inhomogeneity to model a stable local source of instability. The wave propagation and spreading of seizure activity is then a self-organised process as long as the local source is switched on. A key modification of the original model consists in the addition of a variable representing a diffusing species and of variables to feedback the effect of that diffusing species on local cortical activity.

A problem here is that while the assumption that the diffusing species is the extracellular potassium level is a testable hypothesis, this hypothesis is not tested. It is therefore premature to assume that the qualitative similarity of the results support the mechanistic assumptions.

R3.A6. We agree with the Reviewer that, in these patients, we cannot test the specific hypothesis that extracellular potassium supports human seizure. We note that this hypothesis is consistent with previous work in animal models of seizure. We now make this clear in the revised manuscript, as follows:

“Dramatic changes in many extracellular ions occur during seizure⁵³, including increases in extracellular potassium concentration ($[K^+]_o$), which impact neural dynamic and have been proposed as important to seizure activity⁵². The field model proposed here implements these existing concepts, as well as the observation that increases in $[K^+]_o$ act to close gap junctions, through an acidification of the extracellular environment. As the simulated seizure progresses, the increased excitability of the neural populations,

and the reduced coordination of inhibitory cells through a loss of gap junctions, supports the emergence of traveling waves. *Here the hypothesized role of $[K^+]_o$ could not be tested directly in the human patients; future research that incorporates clinically approved methods to detect extracellular ion concentrations would facilitate such a test.*"

R3.Q7. It has been argued previously that rather than picking a single mechanism and adjusting the model parameters until they yield results that resemble the data, testing of alternatives might be a better strategy, Wang et al PLoS Comp Biology (2014). Specifically, the model used in the MS allows for direct implementation of a null hypothesis (e.g. the source of the waves is created by random perturbations of an excitable patch of tissue) which is not done in the current MS.

R3.A7. We agree with this important point, and have updated the manuscript to provide a direct implementation of a null hypothesis in the model. More specifically, we simulate the scenario in which the cortical source of seizure (i.e., the excitable patch of tissue) appears randomly over the two-dimensional space. We find that, under this null hypothesis, the model produces traveling waves with significantly lower directional consistency than that observed in the *in vivo* data or the original model. We mention this result in the revised manuscript as follows:

*"... The direction consistency increases during seizure at both spatial scales ($p < 5 \times 10^{-3}$, Figure 5e), while the velocities simulated are consistent with the *in vivo* values; both range between 50-400 mm/s (Figure 5f). We note that, if instead the source of increased excitability appears at random spatial locations over time, then the direction consistency during seizure is significantly smaller ($p < 0.005$, t-test) in this model (mean 0.47, standard deviation 0.22, during the late seizure interval) compared to the *in vivo* data (mean 0.81, standard deviation 0.16 during the late seizure interval; Supplementary Figure 2 and Figure 6b). Finally, the source directions align during seizure ..."*

In addition, please see Supplementary Figure 2

Also, in the revised manuscript, we now reference [Wang et al., PLoS Comp Biology 2014] when discussing two alternative scenarios of seizure onset:

"We conclude that the two scenarios, both of which are compatible with many aspects of the observed seizure activity, possess particular distinguishing features in terms of their mechanisms, dynamics, and response to stimulation⁶⁹. ..."

F. Suggested improvements: experiments, data for possible revision

R3.Q8. The current MS implements a specific model with a fixed local source and proposes this as an alternative to the previously published proposal of a moving source. I think this needs addressing:

1 The same type of data analysis as previously published needs to be done to check whether the current data sets are of the same or of a different type. If closely related, a combination of the existing data sets could be used to improve the statistics.

R3.A8. In the revised manuscript, we now include a more detailed comparison of these two alternative proposals. Please see R2.A5 for a detailed description of this new material.

In addition, we note here that previous work in [Wagner et al., Neuroimage 2015], which includes some authors of this submission, has repeated the data analysis proposed in [Schevon et al., Nat Comm 2011] on the patients studied in this manuscript. In contrast to [Schevon et al., Nat Comm 2011], the authors of [Wagner et al., Neuroimage 2015] do not find propagation characteristics across seizure that have any obvious correlation with the presence of an ictal wavefront; no clear correlation was seen between the presence of an ictal wavefront and the speed or directionality. That was the case despite the fact that multiunit activity recorded in different sites of the microelectrode array showed clear peaks during the seizure evolution, a hypothesized feature for recruited ictal core areas. This difference in the results from the two groups indicates that the two data sets may reflect different types of seizure and variations across patients.

Nevertheless, we agree with the Reviewer that a combination of the existing data sets from both groups would help resolve these differences. That is something we hope to pursue.

R3.Q9. 2 The mathematical model should implement mechanisms based on both alternatives and check what analysis features are best to distinguish between the two. Also, there is the possibility that both mechanisms might be found in a heterogeneous patient cohort.

R3.A9. As suggested by the Reviewer, we have implemented a mathematical model that simulates the ictal wavefront. We show this new result in the revised Figure 6, and discuss it in the last two paragraphs of Results. Please see R2.A5 for details.

We also mention in the revised manuscript the possibility that both mechanisms might be found in a heterogeneous patient cohort. Please see the revised Discussion in R2.A5, where we state:

“...We note that these two procedures, which exceed the immediate scope of the current study, may reveal that both scenarios occur in a heterogeneous patient cohort⁶⁹.”

R3.Q10. 3 One wonders why this specific model (reference 46) was chosen. It was originally proposed to explain cortical dynamics under anesthesia, whereas numerous

alternative models exist for the description of cortical seizure activity. Specifically, no comment is made on the special feature of this model, namely the vicinity of both a Hopf and a Turing instability, and a parameter region of coexistence of the two. The simulations shown in the MS seem to indicate that the model operates under similar conditions and that this model feature might be a key prerequisite of the observed dynamics (rather than the specific choice of the diffusing extracellular substance).

R3.A10. To provide additional motivation for the model choice, we now state in the revised manuscript that earlier versions of this model have been applied to simulate numerous different scenarios, including seizure.

“... here we focus on the formulation originally proposed in^{39,40} and extended in⁴¹ to simulate the effects of anesthesia. We choose this formulation because it has been successfully extended and interpreted in numerous way, including to study sleep^{42–44}, cognitive states⁴⁵, the effects of anesthesia^{46,47}, and seizure^{31,48,49}.”

The Reviewer is correct that the model implemented here operates under similar conditions to the model defined in [Steyn-Ross et al., PRX, 2013]. Indeed, nearly all of the model parameters are chosen to match those in [Steyn-Ross et al., PRX, 2013], and we observe similar spatiotemporal dynamics as those illustrated in [Steyn-Ross et al., PRX, 2013]. We therefore agree with the Reviewer that the underlying dynamical mechanisms (i.e., the existence of both a Hopf bifurcation and a Turing instability) are critical to producing the simulated spatiotemporal dynamics.

The Reviewer makes a good point that the observed sequence of spatiotemporal dynamics described in this manuscript may occur in other ways. We mention this important point in the revised manuscript as follows:

“... Second, the computational model implements numerous simplifications. The cortex is not a two-dimensional sheet with uniform connectivity, homogenous parameters, and only two cell populations. We induce a sequence of spatiotemporal patterns in the model through a slow change in a variable representative of the concentration of extracellular potassium; slow changes in other model variables may produce similar sequences. We choose to focus on the concentration of extracellular potassium because changes in this ion concentration have been proposed as an important component of seizure^{52–57} ...”

G. References: appropriate credit to previous work?

R3.Q11. Reference to paper by Wang et al PLoS Comp Biology (2014) should be included.

R3.A11. We have updated the manuscript to include this reference in the Discussion as follows:

“We conclude that the two scenarios, both of which are compatible with many aspects of the observed seizure activity, possess particular distinguishing features in terms of their mechanisms, dynamics, and response to stimulation⁶⁹.”

R3.Q12. Reference 25 should be discussed with respect to the outcomes concerning source of spreading wave activity.

R3.A12. As suggested by this Reviewer and the other two Reviewers, we have updated the Results and the Discussion to provide a more detailed comparison of this work and reference 25; please see R2.A5 for details. In addition, as suggested by the Reviewer, we have updated the model to simulate the “ictal wavefront” scenario proposed in reference 25; again, please see the new Figure 6 and R2.A5 for details.

H. Clarity and context: lucidity of abstract/summary, appropriateness of abstract, introduction and conclusions

R3.Q13. MS is well-written. Title and Abstract: I am not sure I fully understand the repeated mention of "couple" or "coupling" across spatial scales. Coherence is a linear measure of amplitude similarity in the frequency domain. So the increase in coherence between scales presumably means an increase in the length of spatial order of the cortical dynamics but not necessarily a "coupling" between the spatial scales.

R3.A13. As recommended by the Reviewer, to clarify further the meaning of “coupling” between spatial scale, we have updated the Abstract to read:

*“We show that during **seizure large-scale neural populations spanning centimeters of cortex coordinate with small neural groups spanning cortical columns**, and provide evidence that rapidly propagating waves of activity underlie this increased inter-scale coupling.”*

R3.Q14. In conclusion, the MS is of high quality and warrants an invitation to revise before a final decision is made.

R3.A14. We are glad the Reviewer found the original manuscript of high quality. We believe that the new manuscript - updated to address the three Reviewers’ concerns - has further improved the quality.

Reviewers' Comments:

Reviewer #1 (Remarks to the Author):

In the revised manuscript “Human seizures couple ...” and the associated reply, the authors (Martinet et al.) responded satisfactorily to most of the critiques. As I will outline below, the only exceptions are R1A8 and R1A11 (and associated referral to R2A5).

R1A8

I do not understand the point the authors are trying to make here. The fact that secondary spots continue to appear after removing the cortical source (shown in reviewer Figure 1) supports the potential significance of these emerging new areas of activation.

R1A11 (and associated referral R2A5)

While I appreciate that the authors confirmed that the descending slope of the Gaussian does not play a role in the presented simulations, I am not sure the Supplementary Figure is necessary. It may be sufficient to mention this finding.

I am confused by the author's referral to R2A5 and the new Figure 6. The findings in the current paper do not necessarily contradict findings described in Schevon et al. (2012) and Smith et al. (2016) and can be easily explained by a wave propagating from a source area (called the ictal core by Schevon et al., 2012). Furthermore, the dynamics in Figure 6a do not represent the scenario proposed by Schevon et al. (2012). I am a strong proponent of using models to verify/falsify hypotheses, but this approach does not accomplish the intended goal.

Finally, I want to comment that the multiscale aspect of this study is oversold in this manuscript. The relationship between the data recorded at the Utah Array and ECoG scales are straightforward, which is confirmed by the applied model. The only multiscale observation is that there are no surprising changes in activity when we zoom in on the mesoscale. Having said this, I do think the findings in this study are valuable and will ultimately help us understand seizure dynamics.

References

Evidence of an inhibitory restraint of seizure activity in humans.

Schevon CA, Weiss SA, McKhann G Jr, Goodman RR, Yuste R, Emerson RG, Trevelyan AJ. Nat Commun. 2012;3:1060. doi: 10.1038/ncomms2056.

The ictal wavefront is the spatiotemporal source of discharges during spontaneous human seizures.

Smith EH, Liou JY, Davis TS, Merricks EM, Kellis SS, Weiss SA, Greger B, House PA, McKhann GM 2nd, Goodman RR, Emerson RG, Bateman LM, Trevelyan AJ, Schevon CA.

Nat Commun. 2016 Mar 29;7:11098. doi: 10.1038/ncomms11098.

Wim van Drongelen

Professor

Depts. of Pediatrics, Neurology

Committee on Computational Neuroscience

Technical Director Pediatric Epilepsy Center

Research Director Pediatric Epilepsy Program

Senior Fellow Computation Institute

The University of Chicago

KCBD 4124, 900 E 57th Street, Chicago, IL

Reviewer #2 (Remarks to the Author):

So I was stunned and disheartened to read all of the reviewers' comments and the authors' responses upon hitting the speedbump from Reviewer 1:

“R1.Q1. Line 18: The latest definition of epilepsy (ILAE 2014) does not necessarily require seizures (i.e. plural).”

Which led to the response:

“R1.A1. As suggested, we have updated the text in the revised manuscript and replaced many instances of “seizures” with “seizure”.”

Which led to the Authors' replacing throughout the text 'seizures' by the grammatically and physiologically incomprehensible use of the singular 'seizure'.

Once my reactive depression to having been so out of it that I had not realized that even my very use of the word 'seizures' throughout my working day was obsolete, I went to the ILAE website, and reviewed the 2014 and more recent roadmap definitions that I had tried to avoid reading throughout my professional life:

http://www.ilae.org/Visitors/Centre/Definition_Class.cfm

To my delight, the arm twisting required to force the authors to damage the readability of their incredibly fine work appeared inconsistent with the facts:

“A person is considered to have epilepsy if they meet any of the following conditions:

1. At least two unprovoked (or reflex) seizures occurring greater than 24 hours apart.
2. One unprovoked (or reflex) seizure and a probability of further seizures similar to the general recurrence risk (at least 60%) after two unprovoked seizures, occurring over the next 10 years.
3. Diagnosis of an epilepsy syndrome

o Epilepsy is considered to be resolved for individuals who had an age-dependent epilepsy syndrome but are now past the applicable age or those who have remained seizure-free for the last 10 years, with no seizure medicines for the last 5 years.

In the definition, epilepsy is now called a disease, rather than a disorder. This was a decision of the Executive Committees of the ILAE and the International Bureau for Epilepsy. Even though epilepsy is a heterogeneous condition, so is cancer or heart disease, and those are called diseases. The word "disease" better connotes the seriousness of epilepsy to the public.

Item 1 of the revised definition is the same as the old definition of epilepsy. Item 2 allows a condition to be considered epilepsy after one seizure if there is a high risk of having another seizure. Often, the risk will not precisely be known and so the old definition will be employed, i.e., waiting for a second seizure before diagnosing epilepsy. Item 3 refers to epilepsy syndromes such as benign epilepsy with central-temporal spikes, previously known as benign rolandic epilepsy, which is usually outgrown by age 16 and always by age 21. If a person is past the age of the syndrome, then epilepsy is resolved. If a person has been seizure-free for at least 10 years with the most recent 5 years off all anti-seizure medications, then their epilepsy also may be considered resolved. Being resolved does not guarantee that epilepsy will not return, but it means the chances are small and the person has a right to consider that she or he is free from epilepsy. This is a big potential benefit of the new definition.”

The entire issue is that one could potentially make an epilepsy diagnosis if you had knowledge of the future (like Dr Who), or perhaps a superb seizure predictor, indicating more than a 60% chance (the lawyers must have offered that one), of a second seizure. Then sanity prevailed and they stated: “Often, the risk will not precisely be known and so the old definition will be employed, i.e., waiting for a second seizure before diagnosing epilepsy.”

The fantastic news is that this reviewer is not yet demented. The other good news is that the authors should feel completely free to use proper use of plurals and grammar in this paper, I am sure to the delight of the Editors.

Other than the present s-deficit, I found that the response of the authors was one of the most comprehensive and impressive of any rebuttal and revision I have ever seen. I think that the manuscript is incredibly improved now, and would be perfect with the re-introduction of all of the needed s's.

Reviewer #3 (Remarks to the Author):

The revision has clarified points raised in the review. The revision has improved the MS such that I recommend publication.

Reviewer #1 (Remarks to the Author)

Round2.R1.Q1. In the revised manuscript “Human seizures couple ...” and the associated reply, the authors (Martinet et al.) responded satisfactorily to most of the critiques. As I will outline below, the only exceptions are R1A8 and R1A11 (and associated referral to R2A5).

Round2.R1.A1. We are glad the Reviewer found the majority of the responses satisfactory. Please find below our responses to the two exceptions. We again thank the Reviewer for raising these questions. We believe that, by addressing each one, we have further improved the clarity of the manuscript.

Round2.R1.Q2. R1A8. I do not understand the point the authors are trying to make here. The fact that secondary spots continue to appear after removing the cortical source (shown in reviewer Figure 1) supports the potential significance of these emerging new areas of activation.

Round2.R1.A2. We apologize that our initial response was not clear. We agree that the secondary spots do represent new areas of activation, which can subsequently evolve into sources of local traveling waves. In retrospect, our previous response focused too finely on the simulation protocol, rather than on this relevant observation. To address this, we have updated the Results section in the revised manuscript to read:

“Examination of the model’s spatial dynamics reveals a distinct sequence of patterns. Before activation of the wave source, the model exists in a “healthy” state in which cortical activity is small. Upon activation of the wave source, excitability spreads from the wave source over the surface, and an approximately static spatial pattern emerges, consisting of active and inactive regions (Figure 5c, label i) [41,50]. During this interval, the initial focus of activity is not clearly distinguished in the 30 cm by 30 cm plane. As time evolves, the extracellular potassium concentration increases in the active regions, thereby increasing the excitability and reducing the gap junction strength in these regions (Supplementary Figure 1). Eventually, these slow changes induce a transition to an interval of local wave propagation, as many brain regions initiate waves that emerge and collide (Figure 5c, label ii). We note that active regions in the approximately static spatial patterns act as transient, secondary sources for this local wave propagation. As the extracellular potassium continues to spread, these waves become more spatially organized until the initial source becomes the clear origination point of all waves, and all transient secondary sources of local wave propagation vanish (Figure 5c, label iii). In this model, the slow evolution of the extracellular potassium concentration navigates the dynamics between different spatiotemporal stages. ~~Reducing the source excitability eliminates the wave source and the cortical sheet returns to quiescence (Figure 5c, label iv). We note that reducing the source excitability earlier in the seizure, after the emergence of the spatial patterns in Figure 5, label i, produces a similar sequence of spatiotemporal states. However, in this case, the traveling waves that appear late in the seizure lack a fixed source, and an approximately simultaneous seizure termination across space—as observed in the human recordings—would require a more global intervention, such as reducing the excitability of the entire cortical sheet.~~”

Round2.R1.Q3. R1A11 (and associated referral R2A5). While I appreciate that the authors confirmed that the descending slope of the Gaussian does not play a role in the

presented simulations, I am not sure the Supplementary Figure is necessary. It may be sufficient to mention this finding.

Round2.R1.A3. We think the Reviewer is referring to R1.A12 (not R1.A11), where we discuss the descending slope of the Gaussian. To address this comment, we have eliminated Supplementary Figure 4 from the revised manuscript, and now only mention this finding in the text, as follows:

“(ii) We included a model of depolarization block, so that the firing rate of each population approaches zero as the voltage exceeds -20 mV; i.e., we apply a Gaussian activation function, rather than the standard sigmoid function [78]. Some recent observations from human seizing cortex suggest an important role for depolarization block [78,79], while others suggest this role may depend on the type of seizure [23,80]. We do not find a critical role for depolarization block in the large amplitude, low frequency dynamics simulated here (~~Supplementary Figure 4~~). However, to simulate the low amplitude, high frequency rhythms commonly observed at seizure onset would require inclusion of additional mechanisms, for example an additional cell population, in which depolarization block may serve an important role.”

Round2.R1.Q4. I am confused by the author’s referral to R2A5 and the new Figure 6. The findings in the current paper do not necessarily contradict findings described in Schevon et al. (2012) and Smith et al. (2016) and can be easily explained by a wave propagating from a source area (called the ictal core by Schevon et al., 2012). Furthermore, the dynamics in Figure 6a do not represent the scenario proposed by Schevon et al. (2012). I am a strong proponent of using models to verify/falsify hypotheses, but this approach does not accomplish the intended goal.

Round2.R1.A4. We thank the Reviewer for helping us clarify this important point. We fully agree that many similarities exist between our results and those in (Smith et al., 2016 and Schevon et al., 2012), with a few differences. Our goal has been to explain accurately these differences, and we did not completely succeed in the previous version of the manuscript. In the resubmission, we have incorporated the Reviewer’s suggestions, which we believe have further improved the comparisons of our work with these two papers. We agree with the Reviewer that the results we report in this manuscript do not contradict the findings described in (Schevon et al. 2012). To make this point clear, we now more carefully distinguish the work in (Schevon et al. 2012) from that group’s later publication (Smith et al., 2016). Accordingly, we have updated the Results section of the revised manuscript to read:

(NOTE: In text below, [25] = Smith et al., 2016, and [26] = Schevon et al., 2012).

“Related scenarios of seizure evolution have been proposed [25,26]. The in vivo activity and simulated data described above are consistent with a small territory of increased activity – an ictal core – that produces widely and rapidly distributed low-frequency (2-50 Hz) fields extending well beyond the ictal core, over broad, multilobar regions [26]. These low-frequency fields travel as waves, which consist of fast-moving synaptic potentials, and produce the large amplitude electroencephalogram (EEG) signature of seizures over broad cortical areas. In another scenario, the seizing territory expands as a slowly

advancing, sharply demarcated, narrow (< 2 mm) band of multiunit firing, termed the ictal wavefront [25]. Traveling waves arise behind the ictal wavefront as it slowly and radially expands across the cortex. ~~These traveling waves, which consist of fast-moving synaptic potentials, produce the low-frequency, large amplitude electroencephalogram (EEG) signature of seizures over broad areas of cortex.~~ This scenario benefits from both clinical [25,26] and experimental [60] observations.

We have also updated the Figure 6 caption and the Discussion to more clearly distinguish the scenarios considered. We now emphasize that the simulations in Figure 6 represent the expanding ictal wavefront described in (Smith et al., 2016). In the revised manuscript, we are also now more careful to indicate the appropriate reference – either (Schevon et al., 2012) or (Smith et al., 2016) – depending on the context. All of these changes resulted in small edits to the manuscript that are indicated in the track-changes version of the resubmission.

Finally, we believe that the scenario in Figure 6 is consistent with the proposal in (Smith et al., 2016). In (Smith et al., 2016), the authors propose that, “*the seizing territory is led by a slowly advancing, sharply demarcated, narrow (<2 mm) band of continuous (tonic) multiunit firing, termed the ictal wavefront. Behind this wavefront, in the seizing territory, there are synchronized rhythmic discharges that give rise to the classic EEG signature of seizures over broad areas of cortex.*” [Page 2, 1st column of Smith et al., 2016]. The ictal discharges “*propagate away from the ictal wavefront ... as it slowly and radially expands across the two-dimensional cortical sheet.*” [Page 3, 2nd column of Smith et al., 2016]. In this way, “*the migrating ictal wavefront is the primary source of ictal activity.*” [Page 6, 2nd column of Smith et al., 2016]. This is the scenario we attempt to capture in Figure 6 and the related text. The authors provide a cartoon illustration of this scenario in Figure 8 of (Smith et al., 2016), which motivated the illustrations in Figure 6c,d of our manuscript.

Round2.R1.Q5. Finally, I want to comment that the multiscale aspect of this study is oversold in this manuscript. The relationship between the data recorded at the Utah Array and ECoG scales are straightforward, which is confirmed by the applied model. The only multiscale observation is that there are no surprising changes in activity when we zoom in on the mesoscale. Having said this, I do think the findings in this study are valuable and will ultimately help us understand seizure dynamics.

Round2.R1.A5. Thank you. We agree that the simplest hypothesis is that field activity propagates in the same direction across spatial scales, which we observe in the data analyzed here. We are glad the Reviewer found the results valuable and helpful to the understanding of seizure dynamics.

References

Evidence of an inhibitory restraint of seizure activity in humans. Schevon CA, Weiss SA, McKhann G Jr, Goodman RR, Yuste R, Emerson RG, Trevelyan AJ. Nat Commun. 2012;3:1060. doi: 10.1038/ncomms2056.

The ictal wavefront is the spatiotemporal source of discharges during spontaneous human seizures. Smith EH, Liou JY, Davis TS, Merricks EM, Kellis SS, Weiss SA, Greger B, House PA, McKhann GM 2nd, Goodman RR, Emerson RG, Bateman LM, Trevelyan AJ, Schevon CA. Nat Commun. 2016 Mar 29;7:11098. doi: 10.1038/ncomms11098.

Wim van Drongelen

Professor

Depts. of Pediatrics, Neurology

Committee on Computational Neuroscience

Technical Director Pediatric Epilepsy Center

Research Director Pediatric Epilepsy Program

Senior Fellow Computation Institute

The University of Chicago

KCBD 4124, 900 E 57th Street, Chicago, IL

Reviewer #2 (Remarks to the Author)

Round2.R2.Q1. So I was stunned and disheartened to read all of the reviewers' comments and the authors' responses upon hitting the speed bump from Reviewer 1:

“R1.Q1. Line 18: The latest definition of epilepsy (ILAE 2014) does not necessarily require seizures (i.e. plural).”

Which led to the response:

“R1.A1. As suggested, we have updated the text in the revised manuscript and replaced many instances of “seizures” with “seizure”.”

Which led to the Authors' replacing throughout the text 'seizures' by the grammatically and physiologically incomprehensible use of the singular 'seizure'.

Once my reactive depression to having been so out of it that I had not realized that even my very use of the word 'seizures' throughout my working day was obsolete, I went to the ILAE website, and reviewed the 2014 and more recent roadmap definitions that I had tried to avoid reading throughout my professional life:

http://www.ilae.org/Visitors/Centre/Definition_Class.cfm

To my delight, the arm twisting required to force the authors to damage the readability of their incredibly fine work appeared inconsistent with the facts:

“A person is considered to have epilepsy if they meet any of the following conditions:

1. At least two unprovoked (or reflex) seizures occurring greater than 24 hours apart.

2. One unprovoked (or reflex) seizure and a probability of further seizures similar to the general recurrence risk (at least 60%) after two unprovoked seizures, occurring over the next 10 years.

3. Diagnosis of an epilepsy syndrome

o Epilepsy is considered to be resolved for individuals who had an age-dependent epilepsy syndrome but are now past the applicable age or those who have remained seizure-free for the last 10 years, with no seizure medicines for the last 5 years.

In the definition, epilepsy is now called a disease, rather than a disorder. This was a decision of the Executive Committees of the ILAE and the International Bureau for Epilepsy. Even though epilepsy is a heterogeneous condition, so is cancer or heart disease, and those are called diseases. The word "disease" better connotes the seriousness of epilepsy to the public.

Item 1 of the revised definition is the same as the old definition of epilepsy. Item 2 allows a condition to be considered epilepsy after one seizure if there is a high risk of having another seizure. Often, the risk will not precisely be known and so the old definition will be employed, i.e., waiting for a second seizure before diagnosing epilepsy. Item 3 refers to epilepsy syndromes such as benign epilepsy with central-temporal spikes, previously known as benign rolandic epilepsy, which is usually outgrown by age 16 and always by age 21. If a person is past the age of the syndrome, then epilepsy is resolved. If a person has been seizure-free for at least 10 years with the most recent 5 years off all anti-seizure medications, then their epilepsy also may be considered resolved. Being resolved does not guarantee that epilepsy will not return, but it means the chances are small and the person has a right to consider that she or he is free from epilepsy. This is a big potential benefit of the new definition."

The entire issue is that one could potentially make an epilepsy diagnosis if you had knowledge of the future (like Dr Who), or perhaps a superb seizure predictor, indicating more than a 60% chance (the lawyers must have offered that one), of a second seizure. Then sanity prevailed and they stated: "Often, the risk will not precisely be known and so the old definition will be employed, i.e., waiting for a second seizure before diagnosing epilepsy."

The fantastic news is that this reviewer is not yet demented. The other good news is that the authors should feel completely free to use proper use of plurals and grammar in this paper, I am sure to the delight of the Editors.

Round2.R2.A1. In an effort to balance the opinions of Reviewer 1 and 2, we have updated the manuscript in those places where "seizures" is grammatically appropriate.

Other than the present s-deficit, I found that the response of the authors was one of the most comprehensive and impressive of any rebuttal and revision I have ever seen. I think that the manuscript is incredibly improved now, and would be perfect with the re-introduction of all of the needed s's.

Round2.R2.A2. We thank the Reviewer - and all the Reviewers - for the useful and insightful comments, which were necessary to improve the manuscript.

Reviewer #3 (Remarks to the Author)

Round2.R3.Q1. The revision has clarified points raised in the review. The revision has improved the MS such that I recommend publication.

Round2.R3.A1. We are glad the Reviewer found the manuscript improved and suitable for publication.